

# A Bayesian Consistent Dual Ensemble Kalman Filter for State-Parameter Estimation in Subsurface Hydrology

B. Ait-El-Fquih [1], M.E. Gharamti [1,2], and I. Hoteit [1]

[1]Department of Earth Sciences and Engineering, King Abdullah University of Science and Technology (KAUST), Thuwal 23955-6900, Kingdom of Saudi Arabia.
[2]Mohn-Sverdrup Center for Global Ocean Studies and Operational Oceanography, Nansen Environmental and Remote Sensing Center (NERSC), Bergen 5006, Norway.

*Correspondence to:* I. Hoteit (ibrahim.hoteit@kaust.edu.sa)

**Abstract.** Ensemble Kalman filtering (EnKF) is an efficient approach to addressing uncertainties in subsurface groundwater models. The EnKF sequentially integrates field data into simulation models to obtain a better characterization of the model's state and parameters. These are generally estimated following joint and dual filtering strategies, in which, at each assimilation cycle, a forecast step by

the model is followed by an update step with incoming observations. The Joint-EnKF directly updates the augmented state-parameter vector while the Dual-EnKF empirically employs two separate filters, first estimating the parameters and then estimating the state based on the updated parameters. To develop a Bayesian consistent dual approach and improve the state-parameters estimates and their consistency, we propose in this paper a one-step-ahead (OSA) smoothing formulation of the

state-parameter Bayesian filtering problem from which we derive a new dual-type EnKF; the Dual-EnKF$_{OSA}$. Compared with the standard Dual-EnKF, it imposes a new update step to the state, which is shown to enhance the performance of the dual approach with almost no increase in the computational cost. Numerical experiments are conducted with a two-dimensional synthetic groundwater aquifer model. Assimilation experiments are performed to assess the performance and robustness of

the proposed Dual-EnKF$_{OSA}$, and to evaluate its results against those of the Joint- and Dual-EnKFs. The proposed scheme is able to successfully recover both the hydraulic head and the aquifer conductivity, further providing reliable estimates of their uncertainties. It is further found more robust to different assimilation settings, such as the spatial and temporal distribution of the observations, and the level of noise in the data. Based on our experimental setups, it yields up to 25% more accurate

state and parameters estimates than the joint and dual approaches.

## 1  Introduction

In modern hydrology research, uncertainty quantification studies have focused on field applications, including surface and subsurface water flow, contaminant transport, and reservoir engineering. The



motivations behind these studies were driven by the uncertain and stochastic nature of hydrological
systems. For instance, surface rainfall-runoff models that account for soil moisture and streamflows
are subject to many uncertain parameters, such as free and tension water storage content, water
depletion rates, and melting threshold temperatures (Samuel et al., 2014). Groundwater flow mod-
els, on the other hand, depend on our knowledge of spatially variable aquifer properties, such as
porosity and hydraulic conductivity, which are often poorly known (Chen and Zhang, 2006; Hen-
dricks Franssen and Kinzelbach, 2008). In addition, contaminant transport models that investigate
the migration of pollutants in subsurface aquifers are quite sensitive to reaction parameters like
sorption rates, radioactive decay, and biodegradation (Gharamti and Hoteit, 2014; Gharamti et al.,
2014b). To this end, it is important to study the variability of hydrological parameters and reduce
their associated uncertainties in order to obtain reliable simulations. To achieve this goal, hydrolo-
gists have resorted to various inverse and Monte Carlo-based statistical techniques with the standard
procedure of pinpointing parameter values that, when integrated with the simulation models, allow
some system-response variables (e.g., hydraulic head, solute concentration) to fit given observa-
tions (Vrugt et al., 2003; Valstar et al., 2004; Alcolea et al., 2006; Feyen et al., 2007; Franssen and
Kinzelbach, 2009; Zhou et al., 2014). Recently, sequential data assimilation techniques, such as the
particle filter (PF), has been proposed to handle any type of statistical distribution, Gaussian or not,
to properly deal with strongly nonlinear systems (Chang et al., 2012). The PF may require, however,
a prohibitive number of particles (and thus model runs) to accurately sample the distribution of the
state and parameters, making this scheme computationally intensive for large-scale hydrological ap-
plications (Doucet et al., 2001; Moradkhani et al., 2005a; Hoteit et al., 2008; Montzka et al., 2011).
This problem has been partially addressed by the popular ensemble Kalman filter (EnKF), which fur-
ther provides robustness, efficiency and non-intrusive formulation (Reichle et al., 2002; Vrugt et al.,
2006; Zhou et al., 2011; Gharamti et al., 2013; Panzeri et al., 2014; Crestani et al., 2013; McMillam
et al., 2013; Erdal and Cirpka, 2015) to tackle the state-parameter estimation problem.

The EnKF is a filtering technique that is relatively simple to implement, even with complex non-
linear models, requiring only an observation operator that maps the state variables from the model
space into the observation space. Compared with traditional inverse and direct optimization tech-
niques, which are generally based on least-squares-like formulations, the EnKF has the advantage
of being able to account for model errors that are not only present in the uncertain parameters but
also in the model structure and inputs, such as external forcings (Hendricks Franssen and Kinzel-
bach, 2008). In addition, and because of its sequential formulation, it does not require storing all
past information about the states and parameters, leading to consequent savings in computational
cost (McLaughlin, 2002; Gharamti et al., 2014b).

The EnKF is widely used in surface and subsurface hydrological studies to tackle state-parameters
estimation problems (Zhou et al., 2014; Panzeri et al., 2014). Two approaches are usually considered
based on the joint and the dual estimation strategies. The standard joint approach concurrently esti-





mates the state and the parameters by augmenting (in the same vector) the state variables with the unknown parameters, that do not vary in time. The parameters could also be set to follow an artificial evolution (random walk) before they get updated with incoming observations (Wan et al., 1999). One of the early applications of the Joint-EnKF to subsurface groundwater flow models was carried

by Chen and Zhang (2006). In their study, a conceptual subsurface flow system was considered and ensemble filtering was performed to estimate the transient pressure field alongside the hydraulic conductivity. In a reservoir engineering application, Nævdal et al. (2005) considered a two-dimensional North Sea field model and considered the joint estimation problem of the dynamic pressure and saturation on top of the static permeability field. Groundwater contamination problems were also

tackled using the Joint-EnKF (e.g., Li et al., 2012; Gharamti and Hoteit, 2014), in which the hydraulic head, contaminant concentration and spatially variable permeability and porosity parameters were estimated for coupled groundwater flow and contaminant transport systems.

Several studies argued that the Joint-EnKF may suffer from important inconsistencies between the estimated state and parameters that could degrade the filter performance, especially with large-

dimensional and strongly nonlinear systems (e.g., Moradkhani et al., 2005b; Chen and Zhang, 2006; Wen and Chen, 2007). One classical approach that has been proposed to tackle this issue is the so-called dual filter which separately updates the state and parameters using two interactive EnKFs, one acting on the state and the other on the parameters (Moradkhani et al., 2005b). The Dual-EnKF has been applied to streamflow forecasting problems using rainfall-runoff models (e.g., Lü

et al., 2013; Samuel et al., 2014), subsurface contaminant (e.g., Tian et al., 2008; Lü et al., 2011; Gharamti et al., 2014b) and compositional flow models (e.g., Phale and Oliver, 2011; Gharamti et al., 2014a), to cite but a few. Gharamti et al. (2014a) concluded that the dual scheme provides more accurate state and parameters estimates than the joint scheme when implemented with large enough ensembles. In terms of complexity, however, the dual scheme requires integrating the filter ensemble

twice with the numerical model at every assimilation cycle, and is therefore computationally more demanding. In related works, Gharamti et al. (2013) extended the dual filtering scheme to tackle the state estimation problem of one-way coupled models, and to the framework of Hybrid-EnKF (Gharamti et al., 2014b).

The dual filter has been basically introduced as a heuristic scheme and is not consistent with the

Bayesian filtering framework (Hendricks Franssen and Kinzelbach, 2008). A first attempt to build a Bayesian consistent dual-like filter was recently proposed by Gharamti et al. (2015) in which a new Joint-EnKF scheme was derived following the one-step-ahead (OSA) smoothing formulation of the Bayesian filtering problem. The new joint scheme reverses the order of the measurement-update and the forecast- (or time-) update, leading to two Kalman-like update steps based on the current

observations; one for state smoothing and one for parameters updating.

Motivated by the promising results of Gharamti et al. (2015), we follow here the same OSA smoothing formulation to derive a new Dual-EnKF, which we refer to as the Dual-EnKF$_{OSA}$ here-



after, generalizing the joint scheme of Gharamti et al. (2015) and, in particular, the standard Dual-EnKF. Our goal being to derive a new Dual-EnKF-like algorithm that retains the separate formulation of the state and parameters update steps, without violating the Bayesian filtering formulation of the state-parameter estimation problem. This not only allows us to derive a more general and efficient dual-like filtering scheme at practically no increase in the computational cost, but also to explicitly put in context the conditions under which the (heuristic) steps of the standard Dual-EnKF can be derived in a Bayesian setting. Synthetic numerical experiments based on a groundwater flow model and estimating the hydraulic head and the conductivity parameter field, are conducted to assess the performances of the proposed Dual-EnKF$_{\mathrm{OSA}}$ and to compare them against the Joint- and the Dual-EnKFs, which we consider as references to evaluate the behavior of the Dual-EnKF$_{\mathrm{OSA}}$. The numerical results suggest that the proposed scheme is beneficial in terms of estimation accuracy compared to the two standard joint and dual schemes, and is more robust to various experiments setting and observations scenarios.

The rest of the paper is organized as follows. Section 2 reviews the standard Joint- and Dual-EnKF strategies. The Dual-EnKF$_{\mathrm{OSA}}$ is derived in Section 3 and its relation with the Joint- and Dual-EnKFs is discussed. Section 4 presents a conceptual groundwater flow model and outlines the experimental setup. Numerical results are presented and discussed in Section 5. Conclusions are offered in Section 6, followed by an Appendix.

## 2 Joint and dual ensemble Kalman filtering

### 2.1 Problem formulation

Consider a discrete-time state-parameter dynamical system:

$$
\begin{cases}
\mathbf{x}_n &= \ \mathcal{M}_{n-1}(\mathbf{x}_{n-1},\theta) + \eta_{n-1} \\
\mathbf{y}_n &= \qquad \mathbf{H}_n \mathbf{x}_n + \varepsilon_n
\end{cases},
\tag{1}
$$

in which $\mathbf{x}_n \in \mathbb{R}^{N_x}$ and $\mathbf{y}_n \in \mathbb{R}^{N_y}$ denote the system state and the observation at time $t_n$, of dimensions $N_x$ and $N_y$ respectively, and $\theta \in \mathbb{R}^{N_\theta}$ is the parameter vector of dimension $N_\theta$. $\mathcal{M}_n$ is a nonlinear operator integrating the system state from time $t_n$ to $t_{n+1}$, and the observational operator at time $t_n$, $\mathbf{H}_n$, is assumed to be linear for simplicity; the proposed scheme can be easily extended to the nonlinear case[1]. The model process noise, $\eta = \{\eta_n\}_{n\in\mathbb{N}}$, and the observation process noise, $\varepsilon = \{\varepsilon_n\}_{n\in\mathbb{N}}$, are assumed to be independent, jointly independent and independent of $\mathbf{x}_0$ and $\theta$, which, in turn, are assumed to be independent. Let also $\eta_n$ and $\varepsilon_n$ be Gaussian with zero means and covariances $\mathbf{Q}_n$ and $\mathbf{R}_n$, respectively. Throughout the paper, $\mathbf{y}_{0:n} \stackrel{\text{def}}{=} \{\mathbf{y}_0, \mathbf{y}_1, \cdots, \mathbf{y}_n\}$, and $p(\mathbf{x}_n)$ and $p(\mathbf{x}_n|\mathbf{y}_{0:l})$ stand for the prior probability density function (pdf) of $\mathbf{x}_n$ and the posterior pdf of $\mathbf{x}_n$ given $\mathbf{y}_{0:l}$, respectively. All other used pdfs are defined in a similar way.

---

[1]The term $\mathbf{H}_n \mathbf{x}_n^{f,(m)}$ is replaced by $\mathcal{H}_n(\mathbf{x}_n^{f,(m)})$ in (26), and $\mathbf{H}_n \xi_n^{(m)}$ is replaced by $\mathcal{H}_n(\xi_n^{(m)})$ in (34).





We focus on the state-parameter *filtering* problem, say, the estimation at each time, $t_n$, of the state, $\mathbf{x}_n$, as well as the parameters vector, $\theta$, from the history of the observations, $\mathbf{y}_{0:n}$. The standard solution of this problem is the *a posteriori* mean (AM):

$$\mathbb{E}_{p(\mathbf{x}_n|\mathbf{y}_{0:n})}[\mathbf{x}_n] = \int \mathbf{x}_n p(\mathbf{x}_n, \theta|\mathbf{y}_{0:n}) d\mathbf{x}_n d\theta, \tag{2}$$

$$\mathbb{E}_{p(\theta|\mathbf{y}_{0:n})}[\theta] = \int \theta p(\mathbf{x}_n, \theta|\mathbf{y}_{0:n}) d\mathbf{x}_n d\theta, \tag{3}$$

which minimizes the *a posteriori* mean square error. In practice, analytical computation of Eqs. (2) and (3) is not feasible, mainly due to the nonlinear character of the system. The Joint- and Dual-EnKFs have been introduced as efficient schemes to compute approximations of (2) and (3). These algorithms are reviewed in the next section.

### 2.2   The Joint- and Dual- EnKFs

#### 2.2.1   The Joint-EnKF

The key idea behind the Joint-EnKF is to transform the state-parameter system (1) into a classical state-space system based on the augmented state, $\mathbf{z}_n = \left[\mathbf{x}_n^T \, \theta^T\right]^T$, on which the classical EnKF can be directly applied. The new augmented state-space system can be written as:

$$\begin{cases} \mathbf{z}_n &= \widetilde{\mathcal{M}}_{n-1}(\mathbf{z}_{n-1}) + \widetilde{\eta}_{n-1} \\ \mathbf{y}_n &= \widetilde{\mathbf{H}}_n \mathbf{z}_n + \varepsilon_n \end{cases}, \tag{4}$$

in which $\widetilde{\mathcal{M}}_{n-1}(\mathbf{z}_{n-1}) = \begin{bmatrix} \mathcal{M}_{n-1}(\mathbf{z}_{n-1}) \\ \theta \end{bmatrix}$, $\widetilde{\eta}_{n-1} = \left[\eta_{n-1}^T \, \mathbf{0}\right]^T$, $\widetilde{\mathbf{H}}_n = [\mathbf{H}_n \, \mathbf{0}]$, with $\mathbf{0}$ a zero matrix with appropriate dimensions. Starting at time $t_{n-1}$ from an analysis ensemble of size $N_e$, $\{\mathbf{x}_{n-1}^{a,(m)}, \theta_{|n-1}^{(m)}\}_{m=1}^{N_e}$ representing $p(\mathbf{z}_{n-1}|\mathbf{y}_{0:n-1})$, the EnKF uses the augmented state model (1st Eq. of (4)) to compute the forecast ensemble, $\{\mathbf{x}_n^{f,(m)}, \theta_{|n-1}^{(m)}\}_{m=1}^{N_e}$, approximating $p(\mathbf{z}_n|\mathbf{y}_{0:n-1})$. The observation model (2nd Eq. of (4)) is then used to obtain the analysis ensemble, $\{\mathbf{x}_n^{a,(m)}, \theta_{|n}^{(m)}\}_{m=1}^{N_e}$,

at time $t_n$. Let, for an ensemble $\{\mathbf{r}^{(m)}\}_{m=1}^{N_e}$, $\hat{\mathbf{r}}$ denotes its empirical mean and $\mathbf{S_r}$ a matrix with $N_e$-columns whose $m^{\text{th}}$ column is defined as $(\mathbf{r}^{(m)} - \hat{\mathbf{r}})$. The Joint-EnKF steps can be summarized as follows:

- *Forecast step:* The parameters vector members, $\theta_{|n-1}^{(m)}$, are kept invariant, while the state vector members, $\mathbf{x}_{n-1}^{a,(m)}$, are integrated in time through the dynamical model as:

$$\mathbf{x}_n^{f,(m)} = \mathcal{M}_{n-1}\left(\mathbf{x}_{n-1}^{a,(m)}, \theta_{|n-1}^{(m)}\right) + \eta_{n-1}^{(m)}; \quad \eta_{n-1}^{(m)} \sim \mathcal{N}(\mathbf{0}, \mathbf{Q}_{n-1}). \tag{5}$$

An approximation of the forecast state, $\hat{\mathbf{x}}_{n|n-1}$, which is, by definition, the mean of $p(\mathbf{x}_n|\mathbf{y}_{0:n-1})$ (as in (2)), is given by the empirical mean of the forecast ensemble, $\hat{\mathbf{x}}_n^f$. The associated forecast error covariance is estimated as $\mathbf{P}_{\mathbf{x}_n^f} = (N_e - 1)^{-1} \mathbf{S}_{\mathbf{x}_n^f} \mathbf{S}_{\mathbf{x}_n^f}^T$.





- *Analysis step:* Once a new observation is available, all members, $\mathbf{x}_n^{f,(m)}$ and $\theta_{|n-1}^{(m)}$, are updated as in the Kalman filter (KF):

$$\mathbf{y}_n^{f,(m)} = \mathbf{H}_n \mathbf{x}_n^{f,(m)} + \varepsilon_n^{(m)}; \quad \varepsilon_n^{(m)} \sim \mathcal{N}(\mathbf{0}, \mathbf{R}_n), \tag{6}$$

$$\mathbf{x}_n^{a,(m)} = \mathbf{x}_n^{f,(m)} + \mathbf{P}_{\mathbf{x}_n^f, \mathbf{y}_n^f} \underbrace{\mathbf{P}_{\mathbf{y}_n^f}^{-1} \left( \mathbf{y}_n - \mathbf{y}_n^{f,(m)} \right)}_{\mu_n^{(m)}}, \tag{7}$$

$$\theta_{|n}^{(m)} = \theta_{|n-1}^{(m)} + \mathbf{P}_{\theta_{|n-1}, \mathbf{y}_n^f} \cdot \mu_n^{(m)}. \tag{8}$$

The (cross-)covariances in Eqs. (7) and (8) are practically evaluated from the ensembles as:

$$\mathbf{P}_{\mathbf{x}_n^f, \mathbf{y}_n^f} = (N_e - 1)^{-1} \mathbf{S}_{\mathbf{x}_n^f} \mathbf{S}_{\mathbf{y}_n^f}^T, \tag{9}$$

$$\mathbf{P}_{\mathbf{y}_n^f} = (N_e - 1)^{-1} \mathbf{S}_{\mathbf{y}_n^f} \mathbf{S}_{\mathbf{y}_n^f}^T, \tag{10}$$

$$\mathbf{P}_{\theta_{|n-1}, \mathbf{y}_n^f} = (N_e - 1)^{-1} \mathbf{S}_{\theta_{|n-1}} \mathbf{S}_{\mathbf{y}_n^f}^T. \tag{11}$$

The analysis estimates, (2) and (3), and their error covariances, can thus be approximated by the analysis ensemble means, $\hat{\mathbf{x}}_n^a$ and $\hat{\theta}_{|n}$, and covariances $\mathbf{P}_{\mathbf{x}_n^a} = (N_e - 1)^{-1} \mathbf{S}_{\mathbf{x}_n^a} \mathbf{S}_{\mathbf{x}_n^a}^T$ and $\mathbf{P}_{\theta_{|n}} = (N_e - 1)^{-1} \mathbf{S}_{\theta_{|n}} \mathbf{S}_{\theta_{|n}}^T$, respectively. Note that $\mathbf{P}_{\mathbf{x}_n^f, \mathbf{y}_n^f} \mathbf{P}_{\mathbf{y}_n^f}^{-1}$ in (7) represents the Kalman Gain, $\mathbf{P}_{\mathbf{x}_n^f} \mathbf{H}_n^T \left[ \mathbf{H}_n \mathbf{P}_{\mathbf{x}_n^f} \mathbf{H}_n^T + \mathbf{R}_n \right]^{-1}$. This statistical formulation of the Kalman Gain offers more flexibility to deal with nonlinear observational operators (Moradkhani et al., 2005b).

### 2.2.2 The Dual-EnKF

In contrast with the Joint-EnKF, the Dual-EnKF is empirically designed following a conditional estimation strategy, operating as a succession of two EnKF-like filters. First, a (parameter) filter is applied to compute $\{\theta_{|n}^{(m)}\}_{m=1}^{N_e}$ from $\{\mathbf{x}_{n-1}^{a,(m)}, \theta_{|n-1}^{(m)}\}_{m=1}^{N_e}$ based on the following two steps.

- *Forecast step:* The parameters ensemble, $\{\theta_{|n-1}^{(m)}\}_{m=1}^{N_e}$, is kept invariant, while the state samples are integrated in time as in (5) to compute the forecast ensemble, $\{\mathbf{x}_n^{f,(m)}\}_{m=1}^{N_e}$.

- *Analysis step:* As in (6), the observation forecast ensemble $\{\mathbf{y}_n^{f,(m)}\}_{m=1}^{N_e}$ is computed from $\{\mathbf{x}_n^{f,(m)}\}_{m=1}^{N_e}$. This is then used to update the parameters ensemble, $\{\theta_{|n}^{(m)}\}_{m=1}^{N_e}$, following (8).

Another (state) filter is then applied to compute $\{\mathbf{x}_n^{a,(m)}\}_{m=1}^{N_e}$ from $\{\mathbf{x}_{n-1}^{a,(m)}\}_{m=1}^{N_e}$ as well as the new parameter ensemble, $\{\theta_{|n}^{(m)}\}_{m=1}^{N_e}$, again in two steps that can be summarized as follows.

- *Forecast step:* Each member, $\mathbf{x}_{n-1}^{a,(m)}$, is propagated in time with the dynamical model using the updated parameters ensemble:

$$\widetilde{\mathbf{x}}_n^{f,(m)} = \mathcal{M}_{n-1} \left( \mathbf{x}_{n-1}^{a,(m)}, \theta_{|n}^{(m)} \right). \tag{12}$$





- *Analysis step:* As in the parameter filter, $\{\widetilde{\mathbf{y}}_n^{f,(m)}\}_{m=1}^{N_e}$ is computed from $\{\widetilde{\mathbf{x}}_n^{f,(m)}\}_{m=1}^{N_e}$ using (6), which finally yields $\{\mathbf{x}_n^{a,(m)}\}_{m=1}^{N_e}$ as in (7).

To better understand how the Dual-EnKF differs from the Joint-EnKF, we focus on how the analysis members at time $t_n$, namely, $\mathbf{x}_n^{a,(m)}$ and $\theta_{|n}^{(m)}$, are obtained starting from their counterparts at previous time, $\mathbf{x}_{n-1}^{a,(m)}$ and $\theta_{|n-1}^{(m)}$. The parameters members, $\theta_{|n}^{(m)}$, are computed based on the same equation (8) in both algorithms. For the state members, $\mathbf{x}_n^{a,(m)}$, we have:

$$\mathbf{x}_n^{a,(m)} \overset{\text{Joint}-\text{EnKF}}{=} \mathcal{M}_{n-1}\left(\mathbf{x}_{n-1}^{a,(m)},\theta_{|n-1}^{(m)}\right) + \mathbf{P}_{\mathbf{x}_n^f,\mathbf{y}_n^f}\underbrace{\mathbf{P}_{\mathbf{y}_n^f}^{-1}\left(\mathbf{y}_n - \mathbf{y}_n^{f,(m)}\right)}_{\mu_n^{(m)}}, \tag{13}$$

$$\mathbf{x}_n^{a,(m)} \overset{\text{Dual}-\text{EnKF}}{=} \underbrace{\mathcal{M}_{n-1}(\mathbf{x}_{n-1}^{a,(m)},\overbrace{\theta_{|n-1}^{(m)} + \mathbf{P}_{\theta_{|n-1},\mathbf{y}_n^f}\cdot\mu_n^{(m)}}^{\theta_{|n}^{(m)}})}_{\widetilde{\mathbf{x}}_n^{f,(m)}} + \mathbf{P}_{\widetilde{\mathbf{x}}_n^f,\widetilde{\mathbf{y}}_n^f}\underbrace{\mathbf{P}_{\widetilde{\mathbf{y}}_n^f}^{-1}\left(\mathbf{y}_n - \widetilde{\mathbf{y}}_n^{f,(m)}\right)}_{\widetilde{\mu}_n^{(m)}}. \tag{14}$$

For simplicity, we ignore here the process noise term, $\eta_n$, which is commonly applied in geophysics applications. As one can see, the Joint-EnKF updates the state members using one Kalman-like correction (term of $\mu_n^{(m)}$ in (13)), while the Dual-EnKF applies two Kalman-like corrections. More specifically, the Dual-EnKF updates first the parameters members, $\theta_{|n-1}^{(m)}$, as in the Joint-EnKF, leading to $\theta_{|n}^{(m)}$; these are then used to propagate $\mathbf{x}_{n-1}^{a,(m)}$, with the model to provide the "forecast" members $\widetilde{\mathbf{x}}_n^{f,(m)}$. The $\widetilde{\mathbf{x}}_n^{f,(m)}$ are finally updated using a Kalman-like correction (term of $\widetilde{\mu}_n^{(m)}$ in (14)), to obtain the analysis members $\mathbf{x}_n^{a,(m)}$. Such a separation of the update steps was shown to provide more consistent estimates of the parameters, especially for strongly heterogeneous subsurface formations as suggested by Wen and Chen (2007) in their confirming-step EnKF algorithm. The dual-update framework was further shown to provide better performances than the Joint-EnKF, at the cost of increased computational burden (see for instance, Moradkhani et al., 2005b; Samuel et al., 2014; Gharamti et al., 2014a).

### 2.2.3 Probabilistic formulation

Following a probabilistic formulation, the augmented state system (4) can be viewed as a continuous state Hidden Markov Chain (HMC) with transition density,

$$p(\mathbf{z}_n|\mathbf{z}_{n-1}) = p(\mathbf{x}_n|\mathbf{x}_{n-1},\theta)\,p(\theta|\theta) = \mathcal{N}_{\mathbf{x}_n}\left(\mathcal{M}_{n-1}\left(\mathbf{x}_{n-1},\theta\right),\mathbf{Q}_{n-1}\right), \tag{15}$$

and likelihood,

$$p(\mathbf{y}_n|\mathbf{z}_n) = p(\mathbf{y}_n|\mathbf{x}_n) = \mathcal{N}_{\mathbf{y}_n}\left(\mathbf{H}_n\mathbf{x}_n,\mathbf{R}_n\right), \tag{16}$$

where $\mathcal{N}_{\mathbf{v}}(\mathbf{m},\mathbf{C})$ represents a Gaussian pdf of argument $\mathbf{v}$ and parameters $(\mathbf{m},\mathbf{C})$.

One can then easily verify that the Joint-EnKF can be derived from a direct application of two classical results of random sampling (Properties 1 and 2 in Appendix A) on the following classical



generic formulas:

$$p(\mathbf{z}_n|\mathbf{y}_{0:n-1}) = \int p(\mathbf{x}_n|\mathbf{x}_{n-1},\theta)\,p(\mathbf{z}_{n-1}|\mathbf{y}_{0:n-1})\,d\mathbf{x}_{n-1}, \tag{17}$$

$$p(\mathbf{y}_n|\mathbf{y}_{0:n-1}) = \int p(\mathbf{y}_n|\mathbf{x}_n)\,p(\mathbf{x}_n|\mathbf{y}_{0:n-1})\,d\mathbf{x}_n, \tag{18}$$

$$p(\mathbf{z}_n|\mathbf{y}_{0:n}) = \frac{p(\mathbf{z}_n,\mathbf{y}_n|\mathbf{y}_{0:n-1})}{p(\mathbf{y}_n|\mathbf{y}_{0:n-1})}. \tag{19}$$

Eq. (17) refers to a *Markovian* step (or time-update step) and uses the transition pdf, $p(\mathbf{x}_n|\mathbf{x}_{n-1},\theta)$, of the Markov chain, $\{\mathbf{z}_n\}_n$, to compute the forecast pdf of $\mathbf{z}_n$ from the previous analysis pdf. Eq. (19) refers to a *Bayesian* step (or measurement-update step) since it uses the Bayes' rule to update the forecast pdf of $\mathbf{z}_n$ using the current observation $\mathbf{y}_n$. Now, establishing the link between the Joint-EnKF and Eqs. (17)-(19), one can show that Property 1 and Eq. (17) lead to the forecast ensemble of the state (5). Property 1 and Eq. (18) lead to the forecast ensemble of the observations (6). Property 2 and Eq. (19) then provide the analysis ensemble of the state (7) and the parameters (8).

Regarding the Dual-EnKF, the forecast ensemble of the state and observations in the parameter filter can be obtained following the same process as in the Joint-EnKF. This is followed by the computation of the analysis ensemble of the parameters using Property 2 and

$$p(\theta|\mathbf{y}_{0:n}) = \frac{p(\theta,\mathbf{y}_n|\mathbf{y}_{0:n-1})}{p(\mathbf{y}_n|\mathbf{y}_{0:n-1})}. \tag{20}$$

However, in the state filter, the ensemble, $\{\widetilde{\mathbf{x}}_n^{f,(m)}\}_{m=1}^{N_e}$, obtained via Eq. (12) in the forecast step does not represent the forecast pdf, $p(\mathbf{x}_n|\mathbf{y}_{0:n-1})$, since Eq. (12) involves $\theta_{|n}^{(m)}$ rather than $\theta_{|n-1}^{(m)}$. Accordingly, the Dual-EnKF is basically a heuristic algorithm in spite of its proven performance.

### 3   One-step-ahead smoothing-based Dual-EnKF (Dual-EnKF$_{\text{OSA}}$)

The classical (time-update, measurement-update) path (17)-(19) to compute the analysis pdf $p(\mathbf{z}_n|\mathbf{y}_{0:n})$ from $p(\mathbf{z}_{n-1}|\mathbf{y}_{0:n-1})$, is not the only possible one. One may indeed reverse the order the time- and measurement-update steps by involving the OSA smoothing pdf, $p(\mathbf{z}_{n-1}|\mathbf{y}_{0:n})$, between two successive analysis pdfs, $p(\mathbf{z}_{n-1}|\mathbf{y}_{0:n-1})$ and $p(\mathbf{z}_n|\mathbf{y}_{0:n})$. Desbouvries et al. (2011) considered the OSA smoothing-based filtering problem in low-dimensional state-space systems to derive a class of KF- and PF-like algorithms for filtering the state. The more recent work of Lee and Farmer (2014) proposes a number of algorithms to estimate both the system state and the model noise based on a similar strategy. In the context of large-dimensional state-parameters filtering, we show in this section that this leads to a new fully Bayesian consistent dual-like filtering scheme, the Dual-EnKF$_{\text{OSA}}$, which, compared to the standard Dual-EnKF, not only introduces another Kalman-like update of the state but also involves a (new) smoothing step that constraints the state with the future observation. Exploiting the future observation should be particularly beneficial in the context of the EnKF as it includes more information in the estimation process that may help mitigating for the subopti-





mal character of the EnKF-like methods, being formulated under a linear Gaussian framework, and usually implemented with limited ensembles and crude approximate noise statistics.

### 3.1 The one-step-ahead smoothing-based dual filtering algorithm

The analysis pdf, $p(\mathbf{x}_n, \theta | \mathbf{y}_{0:n})$, can be computed from $p(\mathbf{x}_{n-1}, \theta | \mathbf{y}_{0:n-1})$ in two steps:

- *Smoothing step:* The one-step-ahead smoothing pdf, $p(\mathbf{x}_{n-1}, \theta | \mathbf{y}_{0:n})$, is first computed as,

$$p(\mathbf{x}_{n-1}, \theta | \mathbf{y}_{0:n}) \propto p(\mathbf{y}_n | \mathbf{x}_{n-1}, \theta, \mathbf{y}_{0:n-1}) p(\mathbf{x}_{n-1}, \theta | \mathbf{y}_{0:n-1}), \tag{21}$$

with,

$$
\begin{aligned}
p(\mathbf{y}_n | \mathbf{x}_{n-1}, \theta, \mathbf{y}_{0:n-1}) &= \int p(\mathbf{y}_n | \mathbf{x}_n, \mathbf{x}_{n-1}, \theta, \mathbf{y}_{0:n-1}) p(\mathbf{x}_n | \mathbf{x}_{n-1}, \theta, \mathbf{y}_{0:n-1}) d\mathbf{x}_n, \\
&= \int p(\mathbf{y}_n | \mathbf{x}_n) p(\mathbf{x}_n | \mathbf{x}_{n-1}, \theta) d\mathbf{x}_n. \tag{22}
\end{aligned}
$$

Eq. (22) is derived from the fact that in the state-parameter model (1), the observation noise, $\varepsilon_n$, and the model noise, $\eta_{n-1}$, are independent of $(\mathbf{x}_{n-1}, \theta)$ and past observations $\mathbf{y}_{0:n-1}$. The smoothing step (21) is indeed a measurement-update step since given $\mathbf{y}_{0:n-1}$, Eq. (21) translates the computation of the posterior, $p(\mathbf{x}_{n-1}, \theta | \mathbf{y}_n)$, as a normalized product of the prior, $p(\mathbf{x}_{n-1}, \theta)$, and the likelihood, $p(\mathbf{y}_n | \mathbf{x}_{n-1}, \theta)$ (note from (22) that $p(\mathbf{y}_n | \mathbf{x}_{n-1}, \theta, \mathbf{y}_{0:n-1})$ $= p(\mathbf{y}_n | \mathbf{x}_{n-1}, \theta)$).

- *Forecast step:* The smoothing pdf at $t_{n-1}$ is then used to compute the current analysis pdf, $p(\mathbf{x}_n, \theta | \mathbf{y}_{0:n})$, as

$$p(\mathbf{x}_n, \theta | \mathbf{y}_{0:n}) = \int p(\mathbf{x}_n | \mathbf{x}_{n-1}, \theta, \mathbf{y}_{0:n}) p(\mathbf{x}_{n-1}, \theta | \mathbf{y}_{0:n}) d\mathbf{x}_{n-1}, \tag{23}$$

with,

$$p(\mathbf{x}_n | \mathbf{x}_{n-1}, \theta, \mathbf{y}_{0:n}) \propto p(\mathbf{y}_n | \mathbf{x}_n) p(\mathbf{x}_n | \mathbf{x}_{n-1}, \theta), \tag{24}$$

which, in turn, arises from the fact that $\varepsilon_n$ and $\eta_{n-1}$ are independent of $(\mathbf{x}_{n-1}, \theta)$ and $\mathbf{y}_{0:n-1}$ (see smoothing step above). We note here that only the (marginal) analysis pdf of $\mathbf{x}_n$, $p(\mathbf{x}_n | \mathbf{y}_{0:n})$, is of interest since the analysis pdf of $\theta$ has already been computed in the smoothing step. From (24), $p(\mathbf{x}_n | \mathbf{x}_{n-1}, \theta, \mathbf{y}_{0:n}) = p(\mathbf{x}_n | \mathbf{x}_{n-1}, \theta, \mathbf{y}_n)$. Thereby, there is a similarity between Eq. (23) and the forecast step (17) in the sense that (23) can be seen as a forecast step once the observation $\mathbf{y}_n$ is known, *i.e.*, (23) coincides with "(17) given the observation $\mathbf{y}_n$". Accordingly, and without abuse of language, we refer to (23)-(24) as the *forecast* step.

### 3.2 Ensemble Formulation

Since it is not possible to derive the analytical solution of (21)-(24) because of the nonlinear character of the model, $\mathcal{M}(.)$, we use Properties 1 and 2 (see Appendix A) to propose an EnKF-like





formulation, assuming that $p(\mathbf{y}_n, \mathbf{z}_{n-1}|\mathbf{y}_{0:n-1})$ is Gaussian for all $n$. This assumption implies that $p(\mathbf{z}_{n-1}|\mathbf{y}_{0:n-1})$, $p(\mathbf{z}_{n-1}|\mathbf{y}_{0:n})$ and $p(\mathbf{y}_n|\mathbf{y}_{0:n-1})$ are Gaussian.

### 3.2.1 Smoothing step

Starting at time $t_{n-1}$, from an analysis ensemble, $\{\mathbf{x}_{n-1}^{a,(m)}, \theta_{|n-1}^{(m)}\}_{m=1}^{N_e}$, one can use Property 1 in Eq. (22) to sample the observation forecast ensemble, $\{\mathbf{y}_n^{f,(m)}\}_{m=1}^{N_e}$, as

$$\mathbf{x}_n^{f,(m)} = \mathcal{M}_{n-1}(\mathbf{x}_{n-1}^{a,(m)}, \theta_{|n-1}^{(m)}) + \eta_{n-1}^{(m)}, \tag{25}$$

$$\mathbf{y}_n^{f,(m)} = \mathbf{H}_n \mathbf{x}_n^{f,(m)} + \varepsilon_n^{(m)}, \tag{26}$$

with $\eta_{n-1}^{(m)} \sim \mathcal{N}(\mathbf{0}, \mathbf{Q}_{n-1})$ and $\varepsilon_n^{(m)} \sim \mathcal{N}(\mathbf{0}, \mathbf{R}_n)$. Property 2 is then used in Eq. (21) to compute the smoothing ensemble, $\{\mathbf{x}_{n-1}^{s,(m)}, \theta_{|n}^{(m)}\}_{m=1}^{N_e}$, as

$$\mathbf{x}_{n-1}^{s,(m)} = \mathbf{x}_{n-1}^{a,(m)} + \mathbf{P}_{\mathbf{x}_{n-1}^a, \mathbf{y}_n^f} \underbrace{\mathbf{P}_{\mathbf{y}_n^f}^{-1} \left(\mathbf{y}_n - \mathbf{y}_n^{f,(m)}\right)}_{\nu_n^{(m)}}, \tag{27}$$

$$\theta_{|n}^{(m)} = \theta_{|n-1}^{(m)} + \mathbf{P}_{\theta_{|n-1}, \mathbf{y}_n^f} \cdot \nu_n^{(m)}. \tag{28}$$

The (cross-) covariances in equations (27) and (28) are defined and evaluated similarly to (9)-(11).

### 3.2.2 Forecast step

The analysis ensemble, $\{\mathbf{x}_n^{a,(m)}\}_{m=1}^{N_e}$, can be obtained from $\{\mathbf{x}_{n-1}^{s,(m)}, \theta_{|n}^{(m)}\}_{m=1}^{N_e}$ using Property 1 in Eq. (23), once the *a posteriori* transition pdf, $p(\mathbf{x}_n|\mathbf{x}_{n-1}, \theta, \mathbf{y}_n)$, is computed via Eq. (24). Furthermore, one can verify that Eq. (24) leads to a Gaussian pdf:

$$p(\mathbf{x}_n|\mathbf{x}_{n-1}, \theta, \mathbf{y}_n) = \mathcal{N}_{\mathbf{x}_n}\left(\mathcal{M}_{n-1}(\mathbf{x}_{n-1}, \theta) + \widetilde{\mathbf{K}}_n\left(\mathbf{y}_n - \mathbf{H}_n\mathcal{M}_{n-1}(\mathbf{x}_{n-1}, \theta)\right), \widetilde{\mathbf{Q}}_{n-1}\right), \tag{29}$$

with $\widetilde{\mathbf{K}}_n = \mathbf{Q}_{n-1}\mathbf{H}_n^T\left[\mathbf{H}_n\mathbf{Q}_{n-1}\mathbf{H}_n^T + \mathbf{R}_n\right]^{-1}$ and $\widetilde{\mathbf{Q}}_{n-1} = \mathbf{Q}_{n-1} - \widetilde{\mathbf{K}}_n\mathbf{H}_n\mathbf{Q}_{n-1}$. However, when the state dimension, $N_x$, is very large, the computational cost of $\widetilde{\mathbf{K}}_n$ and $\widetilde{\mathbf{Q}}_{n-1}$ (which may be a non-diagonal matrix even when $\mathbf{Q}_{n-1}$ is diagonal) may become prohibitive. One way to avoid this problem is to directly sample from $p(\mathbf{x}_n|\mathbf{x}_{n-1}, \theta, \mathbf{y}_n)$ without explicitly computing this pdf in (29). Let $\{\widetilde{\mathbf{x}}_n^{(m)}(\mathbf{x}_{n-1}, \theta)\}_{m=1}^{N_e}$ denotes an ensemble of samples drawn from $p(\mathbf{x}_n|\mathbf{x}_{n-1}, \theta, \mathbf{y}_n)$. The notation $\widetilde{\mathbf{x}}_n^{(m)}(\mathbf{x}_{n-1}, \theta)$ refers to a function $\widetilde{\mathbf{x}}_n^{(m)}$ of $(\mathbf{x}_{n-1}, \theta)$; similar notations hold for $\widetilde{\xi}_n^{(m)}(.)$ and $\widetilde{\mathbf{y}}_n^{(m)}(.)$ in (30) and (31), respectively. Using Properties 1 and 2, an explicit form of such samples can be obtained as (see Appendix B),

$$\widetilde{\xi}_n^{(m)}(\mathbf{x}_{n-1}, \theta) = \mathcal{M}_{n-1}(\mathbf{x}_{n-1}, \theta) + \eta_{n-1}^{(m)}; \quad \eta_{n-1}^{(m)} \sim \mathcal{N}(\mathbf{0}, \mathbf{Q}_{n-1}), \tag{30}$$

$$\widetilde{\mathbf{y}}_n^{(m)}(\mathbf{x}_{n-1}, \theta) = \mathbf{H}_n \widetilde{\xi}_n^{(m)}(\mathbf{x}_{n-1}, \theta) + \varepsilon_n^{(m)}; \quad \varepsilon_n^{(m)} \sim \mathcal{N}(\mathbf{0}, \mathbf{R}_n), \tag{31}$$

$$\widetilde{\mathbf{x}}_n^{(m)}(\mathbf{x}_{n-1}, \theta) = \widetilde{\xi}_n^{(m)}(\mathbf{x}_{n-1}, \theta) + \mathbf{P}_{\widetilde{\xi}_n, \widetilde{\mathbf{y}}_n}\mathbf{P}_{\widetilde{\mathbf{y}}_n}^{-1}\left[\mathbf{y}_n - \widetilde{\mathbf{y}}_n^{(m)}(\mathbf{x}_{n-1}, \theta)\right], \tag{32}$$

where the (cross)-covariances, $\mathbf{P}_{\widetilde{\xi}_n, \widetilde{\mathbf{y}}_n}$ and $\mathbf{P}_{\widetilde{\mathbf{y}}_n}$, are evaluated from the ensembles $\{\widetilde{\xi}_n^{(m)}(\mathbf{x}_{n-1}, \theta)\}_{m=1}^{N_e}$ and $\{\widetilde{\mathbf{y}}_n^{(m)}(\mathbf{x}_{n-1}, \theta)\}_{m=1}^{N_e}$, similarly to (9)-(11). Now, using Property 1 in Eq. (23), one can compute



an analysis ensemble, $\{\mathbf{x}_n^{a,(m)}\}_{m=1}^{N_e}$, from the smoothing ensemble, $\{\mathbf{x}_{n-1}^{s,(m)}, \theta_{|n}^{(m)}\}_{m=1}^{N_e}$, using the functional form (32). More precisely, we obtain, $\mathbf{x}_n^{a,(m)} = \widetilde{\mathbf{x}}_n^{(m)}(\mathbf{x}_{n-1}^{s,(m)}, \theta_{|n}^{(m)})$, which is equivalent to set $\mathbf{x}_{n-1} = \mathbf{x}_{n-1}^{s,(m)}$ and $\theta = \theta_{|n}^{(m)}$ in (30)-(32).

### 3.2.3 Summary of the Dual-EnKF$_{\text{OSA}}$ algorithm

Starting from an analysis ensemble, $\{\mathbf{x}_{n-1}^{a,(m)}, \theta_{|n-1}^{(m)}\}_{m=1}^{N_e}$, at time $t_{n-1}$, the updated ensemble of both state and parameters at time $t_n$ is obtained with the following two steps:

- *Smoothing step:* The state forecast ensemble, $\{\mathbf{x}_n^{f,(m)}\}_{m=1}^{N_e}$, is first computed by (25), and then used to compute the observation forecast ensemble, $\{\mathbf{y}_n^{f,(m)}\}_{m=1}^{N_e}$, as in (26). This latter is then used to compute the one-step-ahead smoothing ensemble of the state, $\{\mathbf{x}_{n-1}^{s,(m)}\}_{m=1}^{N_e}$, and parameters, $\{\theta_{|n}^{(m)}\}_{m=1}^{N_e}$, based on Eqs. (27) and (28), respectively.

- *Forecast step:* The analysis ensemble of the state $\{\mathbf{x}_n^{a,(m)}\}_{m=1}^{N_e}$ is obtained as:

$$\xi_n^{(m)} = \mathcal{M}_{n-1}\left(\mathbf{x}_{n-1}^{s,(m)}, \theta_{|n}^{(m)}\right) + \eta_{n-1}^{(m)}; \quad \eta_{n-1}^{(m)} \sim \mathcal{N}(\mathbf{0}, \mathbf{Q}_{n-1}), \tag{33}$$

$$\widetilde{\mathbf{y}}_n^{f,(m)} = \mathbf{H}_n \xi_n^{(m)} + \varepsilon_n^{(m)}; \quad \varepsilon_n^{(m)} \sim \mathcal{N}(\mathbf{0}, \mathbf{R}_n), \tag{34}$$

$$\mathbf{x}_n^{a,(m)} = \xi_n^{(m)} + \mathbf{P}_{\xi_n, \widetilde{\mathbf{y}}_n^f} \mathbf{P}_{\widetilde{\mathbf{y}}_n^f}^{-1}(\mathbf{y}_n - \widetilde{\mathbf{y}}_n^{f,(m)}), \tag{35}$$

with $\mathbf{P}_{\xi_n, \widetilde{\mathbf{y}}_n^f} = (N_e - 1)^{-1} \mathbf{S}_{\xi_n} \mathbf{S}_{\widetilde{\mathbf{y}}_n^f}^T$ and $\mathbf{P}_{\widetilde{\mathbf{y}}_n^f} = (N_e - 1)^{-1} \mathbf{S}_{\widetilde{\mathbf{y}}_n^f} \mathbf{S}_{\widetilde{\mathbf{y}}_n^f}^T$.

In contrast with the Dual-EnKF which uses $\theta_{|n}^{(m)}$ and $\mathbf{x}_{n-1}^{a,(m)}$ for computing $\mathbf{x}_n^{a,(m)}$ (see Eq. (14)), the proposed Dual-EnKF$_{\text{OSA}}$ uses $\theta_{|n}^{(m)}$ and the smoothed state members, $\mathbf{x}_{n-1}^{s,(m)}$, which are the $\mathbf{x}_{n-1}^{a,(m)}$ after an update with the current observation, $\mathbf{y}_n$, following (27). Therefore, when including the Kalman-like correction term as well, the observation, $\mathbf{y}_n$, is used three times in the Dual-EnKF$_{\text{OSA}}$ in a fully consistent Bayesian formulation, compared to only twice in the Dual-EnKF. This means that the Dual-EnKF$_{\text{OSA}}$ exploits the observations more efficiently than the Dual-EnKF, which should provide more information for improved and more consistent state and parameters estimates.

Despite the smoothing formulation of the Dual-EnKF$_{\text{OSA}}$, this algorithm obviously addresses the state forecast problem as well. As discussed in the smoothing step above, the (one-step-ahead) forecast members are inherently computed. The $j$-step-ahead forecast member, denoted by $\mathbf{x}_{n+j|n}^{(m)}$ for $j \geq 2$, can be computed following a recursive procedure where, for $\ell = 2, 3, \cdots, j$, one has

$$\mathbf{x}_{n+\ell|n}^{(m)} = \mathcal{M}_{n+\ell-1}(\mathbf{x}_{n+\ell-1|n}^{(m)}, \theta_{|n}^{(m)}) + \eta_{n+\ell-1}^{(m)}, \quad \eta_{n+\ell-1}^{(m)} \sim \mathcal{N}(\mathbf{0}, \mathbf{Q}_{n+\ell-1}). \tag{36}$$

### 3.3 Complexity of the Joint-EnKF, Dual-EnKF, and Dual-EnKF$_{\text{OSA}}$

The computational complexity of the different state-parameter EnKF schemes can be split between the forecast (time-update) step and the analysis (measurement-update) step. The Joint-EnKF requires





$N_e$ model runs (for forecasting the state ensemble) and $N_e$ Kalman corrections (for updating the forecast ensemble). This is practically doubled when using the Dual-EnKF, since the latter requires $2N_e$ model runs and $2N_e$ Kalman corrections; $N_e$ corrections for each of the forecast state ensemble and the forecast parameter ensemble. As presented in the previous section, the Dual-EnKF$_{\text{OSA}}$ smoothes the state estimate at the previous time step before updating the parameters and the state at
the current time. Thus, the Dual-EnKF$_{\text{OSA}}$ requires as many model runs ($2N_e$) as the Dual-EnKF, and an additional $N_e$ correction steps to apply smoothing. In large scale geophysical applications, the correction step of the ensemble members is often computationally not significant compared to the cost of integrating the model in the forecast step. The approximate computational complexity and memory storage for each algorithm are summarized in Table 1. The tabulated complexities for each
method are valid under the assumption that $N_y \ll N_x$, *i.e.*, the number of state variables is much larger than the number of observations. This is generally the case for subsurface flow applications due to budget constraints given the consequent costs needed for drilling and maintaining subsurface wells.

## 4   Numerical experiments

### 4.1   Transient groundwater flow problem

We adopt in this study the subsurface flow problem of Bailey and Baù (2010). The system consists of a two-dimensional (2D) transient flow with an areal aquifer area of 0.5 km$^2$ (Figure 1). Constant head boundaries of 20 m and 15 m are placed on the west and east ends of the aquifer, respectively, with an average saturated thickness, $b$, of 25 m. The north and south boundaries are assumed to be
Neumann with no flow conditions (Figure 1). The mesh is discretized using a cell-centered finite difference scheme with 10 m $\times$ 20 m rectangles, resulting in 2500 elements. The following 2D saturated groundwater flow system is solved:

$$\frac{\partial}{\partial x}\left(T_x \frac{\partial h}{\partial x}\right) + \frac{\partial}{\partial y}\left(T_y \frac{\partial h}{\partial y}\right) = S\frac{\partial h}{\partial t} + q, \tag{37}$$

where $T$ is the transmissivity [L$^2$T$^{-1}$], which is related to the conductivity, $K$, through $T = Kb$, $h$ is
the hydraulic head [L], $t$ is time [T], $S$ is storativity [-], and $q$ denotes the sources as recharge or sinks due to pumping wells [LT$^{-1}$]. Unconfined aquifer conditions are simulated by setting $S = 0.20$ to represent the specific yield. A log-conductivity field is generated using the sequential Gaussian simulation toolbox, GCOSIM3D (Gómez-Hernández and Journel, 1993), with a mean of $-13$ log(m/s), a variance of $1.5$ log(m$^2$/s$^2$), and a Gaussian variogram with a range equal to 250 m in the x-direction
and 500 m in the y-direction (Figure 1).

We consider a dynamically complex experimental setup that is similar to a real-world application and is based on various time-dependent external forcings. The recharge is assumed spatially heterogenous and sampled using the GCOSIM3D toolbox (Gómez-Hernández and Journel, 1993) with



statistical parameters shown in Table 2. Three different pumping wells (PW) are inserted within the aquifer domain and can be seen in Figure 1 (square symbols). From these wells, transient pumping of groundwater takes place with different daily values as plotted in the left panel of Figure 2. The highest pumping rates are associated with PW2 with an average daily rate of $5.935 \times 10^{-7}$ m$^3$/day. Smaller temporal variations in water pumping rates are assigned to PW1 and PW3. Three other monitoring wells (MW1, MW2, MW3) are also placed within the aquifer domain to evaluate the groundwater flow filters estimates. We further assess the prediction skill of the model after data assimilation using a control well (CW) placed in the middle of the aquifer (indicated by a diamond symbol).

Prior to assimilation, a reference run is first conducted for each experimental setup using the prescribed parameters above, and is considered as the truth. We simulate the groundwater flow system over a year-and-a-half period using the classical fourth-order Runge Kutta method with a time step of 12 hours. The initial hydraulic head configuration is obtained after a 2-years model spin-up starting from a uniform 15 m head. Reference heterogenous recharge rates are used in the setup as explained before. The water head changes (in m) after 18 months are displayed with contour lines in the left panel of Figure 3. One can notice larger variations in the water head in the lower left corner of the aquifer domain, consistent with the high conductivity values in that region. The effects of transient pumping in addition to the heterogenous recharge rates are also well observed in the vicinity of the pumping wells.

### 4.2 Assimilation Experiments

To imitate a realistic setting, we impose various perturbations on the reference model and set our goal to estimate the water head and the hydraulic conductivity fields using an imperfect forecast model and perturbed data extracted from the reference (true) run. This experimental framework is known as "twin-experiments". In the forecast model, we perturb both transient pumping and spatial recharge rates. The perturbed recharge field, as compared to the reference recharge in Figure 2, is sampled with different variogram parameters as shown in Table 2. Pumping rates from PW1, PW2 and PW3 are perturbed by adding a Gaussian noise with mean zero and standard deviation equal to 20% of the reference transient rates. The flow field simulated by the forecast (perturbed) model after 18 months is shown in the right panel of Figure 3. Compared to the reference field, there are clear spatial differences in the hydraulic head, especially around the first and second pumping wells.

To demonstrate the effectiveness of the proposed Dual-EnKF$_{\text{OSA}}$, we evaluate its performances against the standard Joint- and Dual-EnKFs under different experimental scenarios. We further conduct a number of sensitivity experiments, changing: (1) the ensemble size, (2) the temporal frequency of available observations, (3) the number of observation wells in the domain, and (4) the measurement error. For the frequency of the observations, we consider 6 scenarios in which hydraulic head measurements are extracted from the reference run every 1, 3, 5, 10, 15, and 30 days. Of course,





$\mathbf{x}_n^{a,(m)}$ is equal to $\mathbf{x}_n^{f,(m)}$ when no observation is assimilated. We also test four different observational networks assuming 9, 15, 25 and 81 wells uniformly distributed throughout the aquifer domain (Figure 3 displays two of these networks; with 9 and 25 wells). We evaluate the algorithms under 9 different scenarios in which the observations were perturbed with Gaussian noise of zero mean and a standard deviation equal to 0.10, 0.15, 0.20, 0.25, 0.30, 0.50, 1, 2 and 3 m. Such measurement

errors, which can be due to instruments errors, conversion of pressure to water head, or piezometer well defects, are typical values (order of centimetres to meters) observed at real hydrologic sites (Post and von Asmuth, 2013).

To initialise the filters, we follow Gharamti et al. (2014a) and perform a 5-years simulation run using the perturbed forecast model starting from the mean hydraulic head of the reference run solution.

Then, we randomly select a set of $N_e$ hydraulic head snapshots to form the initial state ensemble. By doing so, the dynamic head changes that may occur in the aquifer are well represented by the initial ensemble. The corresponding parameters' realizations are sampled with the geostatistical software, GCOSIM3D, using the same variogram parameters of the reference conductivity field but conditioned on two hard measurements as indicated by black crosses in Figure 1. The two data points

capture some parts of the high conductivity regions in the domain, and thus one should expect a poor representation of the low conductivity areas in the initial $\log(K)$ ensemble. This is a challenging case for the filters especially when a sparse observational network is considered. To ensure consistency between the hydraulic heads and the conductivities at the beginning of the assimilation, we conduct a spin-up of the whole state-parameters ensemble for a 6-months period using perturbed

recharge time-series for each ensemble member.

The filter estimates resulting from the different filters are evaluated based on their average absolute forecast errors (AAE) and their average ensemble spread (AESP):

$$AAE \quad = \quad N_x^{-1} N_e^{-1} \sum_{j=1}^{N_e} \sum_{i=1}^{N_x} \left| \mathbf{x}_{j,i}^{f,e} - \mathbf{x}_i^t \right|, \tag{38}$$

$$AESP \quad = \quad N_x^{-1} N_e^{-1} \sum_{j=1}^{N_e} \sum_{i=1}^{N_x} \left| \mathbf{x}_{j,i}^{f,e} - \hat{\mathbf{x}}_i^{f,e} \right|, \tag{39}$$

where $\mathbf{x}_i^t$ is the reference "true" value of the variable at cell $i$, $\mathbf{x}_{j,i}^{f,e}$ is the forecast ensemble value of the variable, and $\hat{\mathbf{x}}_i^{f,e}$ is the forecast ensemble mean at location $i$. AAE measures the estimate-truth misfit and AESP measures the ensemble spread, or the confidence in the estimated values (Hendricks Franssen and Kinzelbach, 2008). We further assess the accuracy of the estimates by plotting the resulting field and variance maps of both hydraulic head and conductivities.



## 5 Results and Discussion

### 5.1 Sensitivity to the ensemble size

We first study the sensitivity of the three algorithms to the ensemble size, $N_e$. In realistic ground-water applications, we would be restricted to small ensembles due to computational limitations. Obtaining accurate state and parameter estimates with small ensembles is thus desirable. We carry the experiments using three ensemble sizes, $N_e = 50$, 100 and 300, and we fix the frequency of the observations to half a day, the number of wells to nine (Figure 3, left observation network) and the measurement error to 0.50 m. We plot the resulting AAE time series of the state and parameters in Figure 4. As shown, the performance of the Joint-EnKF, Dual-EnKF and Dual-EnKF$_{\text{OSA}}$ improves as the ensemble size increases, reaching a mean AAE of 0.161, 0.160, and 0.156 m for $N_e = 300$, respectively. The Joint-EnKF and the Dual-EnKF exhibit similar behaviors, with a slight advantage for the Dual-EnKF. As suggested by (Gharamti et al., 2014a), the Dual-EnKF is generally expected to produce more accurate results only when large enough ensembles are used. We have tested the Joint- and the Dual-EnKFs using 1000 members and found that the Dual-EnKF is around 9% more accurate in term of AAE. The proposed Dual-EnKF$_{\text{OSA}}$ provides the best estimates in all tested scenarios. On average, with changing ensemble size, the Dual-EnKF$_{\text{OSA}}$ leads to about 7% improvement compared with the joint and dual schemes. In terms of the conductivity estimates, the proposed scheme produces more accurate estimates for all three ensemble sizes. At the early assimilation stage, the three schemes seem to provide similar results, but this eventually changes after 6 months beyond which the Dual-EnKF$_{\text{OSA}}$ clearly outperforms the standard schemes.

We furthermore examined the estimated uncertainties about the forecast estimates by computing the average spread of both the hydraulic head and conductivity ensembles. To do this, we evaluated the mean AESP of both variables and tabulated the results for the three ensemble sizes in Table 3. For all schemes and as expected, the spread seems to increase as the ensemble size increases. Compared to the joint and the dual schemes, the Dual-EnKF$_{\text{OSA}}$ retains the smallest mean AESP for all cases, suggesting more confidence in the head and conductivity estimates.

One could also exploit the computed AAE and AESP to assess whether the filters suffer from the inbreeding problem. Filter inbreeding occurs when the variance of the state and parameters ensemble is increasingly reduced over time. This may not only deteriorates the quality of the estimated filter error covariance matrices, but also wrongly suggests more confidence in the forecast and strongly limits the filter update by the incoming observation. One standard test for examining inbreeding is to compute the ratio of the AAE to the AESP (Hendricks Franssen and Kinzelbach, 2008). In a well designed assimilation system (that does not suffer from inbreeding) such a ratio should be close to one; in other words, the AAE and AESP are almost of the same order. Examining Figure 4 and Table 4, the ratio of the AAE to the AESP for the different tested ensemble sizes is, on-average, very close to 1 for all three schemes, as reported in Table 4. This clearly suggests that no





filtering inbreeding issues are encountered in the present setup. This could be due to the imposed stochastic model errors (as described in Section 4), which seems to maintain enough spread in the hydraulic head and conductivity ensembles. Another method for tackling the inbreeding problem is to combine the EnKF with the so-called stochastic moments equations that govern the time evolution
of conditional expectations of the state and parameters as well as the associated covariances, as suggested by Panzeri et al. (2013, 2015).

In terms of computational cost, we note that our assimilation results were obtained using a 2.30 GHz MAC workstation and 4 cores for parallel looping while integrating the ensemble members. The Joint-EnKF is the least intensive requiring 70.61 sec to perform a year-and-a-half assimilation run
using 50 members. The Dual-EnKF and Dual-EnKF$_{OSA}$, on the other hand, require 75.37 and 77.04 sec, respectively. The Dual-EnKF is computationally more demanding than the Joint-EnKF because it includes an additional propagation step of the ensemble members as discussed in Section 3.3. Likewise, the proposed Dual-EnKF$_{OSA}$ requires both an additional propagation step and an update step of the state members. Its computational complexity is thus greater than the joint scheme and
roughly equivalent to that of the Dual-EnKF. Note that in the current setup the cost of integrating the groundwater model is not very significant as compared to the cost of the update step. This is however usually not the case in large-scale hydrological applications.

## 5.2 Sensitivity to the frequency of observations

In the second set of experiments, we test the filters' behavior with different temporal frequency of
observations, i.e., the times at which head observations are assimilated. We implement the three filters with 100 members and use data from nine observation wells perturbed with 0.10 m noise.

Figure 5 plots the mean AAE of the hydraulic conductivity estimated using the three filters for the six different observation frequency scenarios. The Dual- and Joint-EnKFs lead to comparable performances, but the latter performs slightly better when data are assimilated more frequently, i.e.,
every five and three days. The performance of the proposed Dual-EnKF$_{OSA}$, as seen from the plot, is rather good and its estimates are more consistent with data than those computed by the other two filters. The best Dual-EnKF$_{OSA}$ results are obtained when assimilating data every 1, 3, and 5 days. The improvements over the joint and the dual schemes decrease as the frequency of observations in time decreases. The reason for this is related to the nature of the Dual-EnKF$_{OSA}$ algorithm, which
adds a one-step-ahead-smoothing to the analyzed head ensemble members before updating the forecast parameters and the state samples. Therefore, the more data are available, the greater the number of applied smoothing steps, and hence the better the characterization of the state and parameters. To illustrate, the smoothing step of the state ensemble enhances its statistics and eventually provides more consistent state-parameters cross-correlations to better predict the data. When assimilating hy-
draulic head data on a daily basis, the proposed Dual-EnKF$_{OSA}$ leads to about 24% more accurate conductivity estimates than the Joint and Dual-EnKFs.





We also compared the hydraulic head estimates when changing the temporal frequency of observations. Similar to the parameters, the improvements of the Dual-EnKF$_{\text{OSA}}$ algorithm over the other schemes become significant when more data are assimilated over time. Overall, the benefits

of the proposed scheme are more pronounced for the estimation of the parameters because the conductivity values at all aquifer cells are indirectly updated using hydraulic head data, requiring more observations for efficient estimation.

One effective way to evaluate the estimates of the state is to examine the evolution of the reference heads and the forecast ensemble members at various aquifer locations. For this, we plot in Figure 6

the true and the estimated time-series change in hydraulic head at the assigned monitoring wells as they result from the Joint-EnKF, Dual-EnKF and the Dual-EnKF$_{\text{OSA}}$. We use 100 ensemble members and assume the 9 data points are available every five days. At MW1, the performance of the three filters is quite similar and they all successfully reduce the uncertainties and recover the true evolution of the hydraulic head at that location. We note that between the $5^{\text{th}}$ and the $9^{\text{th}}$ month,

the Dual-EnKF seems to underestimate the reference values of the hydraulic head as compared to the other two schemes. At MW2 and MW3, the ensemble spread of all three filters shrinks shortly after the start of assimilation, but remains larger than those at MW1. The proposed Dual-EnKF$_{\text{OSA}}$ efficiently recovers the reference trajectory of MW2 and MW3. The ensemble head values obtained using the Joint- and the Dual-EnKFs at MW2 are less accurate. Furthermore, the Joint and the Dual-

EnKF ensemble members tend to underestimate the reference hydraulic head at MW3 over the first 6 months of assimilation. Beyond this, there is a clear overestimation of the head values, especially by the Dual-EnKF, up to the end of the first year.

### 5.3 Sensitivity to the number of observations

We further examine the robustness of the proposed Dual-EnKF$_{\text{OSA}}$ against the Joint- and Dual-

EnKFs to different numbers of observation wells inside the aquifer domain. We thus compare our earlier estimates resulting from only nine wells, five days observation frequency, and 0.10 m measurement error with a new set of estimates resulting from more dense observational networks with 15, 25 and 81 wells. Figure 7 plots the time-series curves of the AAE as they result from the four observational scenarios for hydraulic head and conductivity. As shown, the behavior of the filters

improves as more data are assimilated. Clearly, the proposed scheme provides the best estimates over the entire simulation window. More precisely, and towards the end of assimilation, the Dual-EnKF$_{\text{OSA}}$ with only nine data points exhibits less forecast errors for conductivity than does the Dual-EnKF with 81 data points. Likewise when assimilating head data from 15 and 25 wells, the proposed algorithm outperforms the Dual-EnKF and yields more accurate hydraulic head estimates

by the end of the simulation window.

To further assess the performance of the filters we analyze the spatial patterns of the estimated fields. To do so, we plot and interpret the ensemble mean of the conductivity as it results from the



three filters using nine observation wells. We compare the estimated fields after 18 months (Figure 8) with the reference conductivity. As can be seen, the Joint- and the Dual-EnKFs exhibit some
overshooting in the southern (low conductivity) and central regions of the domain. In contrast, the Dual-EnKF$_{OSA}$ better delineates these regions and further provides reasonable estimates of the low conductivity area in the northwest part of the aquifer. In general and for all tested schemes, the estimated conductivity field does not capture very well the spatial variability of the reference field. This is due to the large model errors imposed on the recharge and pumping rates during the forecasts.
This limits the efficiency of the assimilation system, especially with the recovery of small scale conductivity structures, but also allows for more straightforward assessment of the different techniques.

### 5.4 Sensitivity to measurement errors

In the last set of sensitivity experiments, we fix the number of wells to nine, the observation fre-
quency to 5 days, and we use different measurement errors to perturb the observations. We plot the results of nine different observational error scenarios in Figure 9 and compare the conductivity estimates obtained using the Joint-EnKF, Dual-EnKF and the Dual-EnKF$_{OSA}$. In general, the performance of the filters appears to degrade as the observations are perturbed with larger degree of noise. All three filters exhibit similar performances with large observational error; i.e., 1, 2 and 3 m.
This can be expected because larger observational errors decrease the impact of data assimilation, and thus the estimation process is reduced to a model prediction only. The plot also suggests that the estimates of the Dual-EnKF$_{OSA}$ with 0.30 m measurement errors are better than those of the Joint- and the Dual-EnKFs with 0.10 m error. With 0.10 m measurement error, the estimate of the Dual-EnKF$_{OSA}$ is also approximately 12% better.
Finally, we investigated the time-evolution of the ensemble variance of the conductivity estimates as they result from the Dual-EnKF and the Dual-EnKF$_{OSA}$ with 0.10 m measurement noise. Spatially, the ensemble variance maps provide insight about the uncertainty reduction due to data assimilation. The initial map (left panel, Figure 10) exhibits zero variance at the sampled two locations and increasing variance away from these locations. The ensemble spread of conductivity field from the
two filters (right panels, Figure 10) after 6 and 18 months is quite small and comparable. The Dual-EnKF$_{OSA}$, however, tends to maintain a larger variance at the edges than the Dual-EnKF, which in turn increases the impact of the observations.

### 5.5 Further assessments of the Dual-EnKF$_{OSA}$ scheme

To further assess the system performance in terms of parameters retrieval, we have integrated the
model in prediction mode (without assimilation) for an additional period of 18 months starting from the end of the assimilation period. We plot in Figure 11, using the final estimates of the conductivity as they result from the three filters (after 18 months), the time evolution of the hydraulic head at the



control well (CW). The ensemble size is set to 100, observation frequency is 1 day, number of data
wells is 25 and measurement noise is 0.5 m. The reference head trajectory at the CW decreases from
17.5 to 16.9 m in the first 2 years, and then slightly increases to 17.2 m in the rest of the year. The
forecast ensemble members of the Joint-EnKF at this CW fail to capture to reference trajectory of
the model. This is due to the large measurement noise imposed on the head data. The Dual-EnKF
performs slightly better and predicts hydraulic head values that are closer to the reference solution.
The performance of the Dual-EnKF$_{OSA}$, as shown, is the closest to the reference head trajectory
and moreover, one of the forecast ensemble members successfully captures the true head evolution.
Similar verification was also conducted at other locations in the aquifer and all resulted in similar
results.

Finally, in order to demonstrate that our results are statistically robust, 10 other test cases with
different reference conductivity and heterogeneous recharge maps were investigated. In each of these
cases, we sampled the reference fields by varying the variogram parameters, such as variance, $x$ and
$y$ ranges, etc. The pumping rates and the initial head configuration among the cases were also altered.
For all 10 test cases, we fixed the ensemble size to 100 and used data from nine observation wells
every 3 days. We set the measurement error to 0.10 m. We plot the resulting conductivity estimates
(mean AAE) from each case in Figure 12. The estimates of the three filters, as shown on the plot,
give a statistical evidence that the proposed scheme always provides more accurate estimates than
the Joint-/Dual-EnKF and is more robust to changing dynamics and experimental setups. Similar
results were obtained for the hydraulic head estimates. Averaging over all test cases, the proposed
scheme provides about 17% more accurate estimates in term of AAE than the standard Joint- and
Dual-EnKFs.

## 6   Conclusions


We presented a one-step-ahead smoothing based dual ensemble Kalman filter (Dual-EnKF$_{OSA}$) for
state-parameter estimation of subsurface groundwater flow models. The Dual-EnKF$_{OSA}$ is derived
using a Bayesian probabilistic formulation combined with two classical stochastic sampling proper-
ties. It differs from the standard Joint-EnKF and Dual-EnKF in the fact that the order of the time-
update step of the state (forecast by the model) and the measurement-update step (correction by the
incoming observations) is inverted. Compared with the Dual-EnKF, this introduces a smoothing step
to the state by future observations, which seems to provide the model, at the time of forecasting,
with better and rather physically-consistent state and parameters ensembles.

We tested the proposed Dual-EnKF$_{OSA}$ on a conceptual groundwater flow model in which we es-
timated the hydraulic head and spatially variable conductivity parameters. We conducted a number
of sensitivity experiments to evaluate the accuracy and the robustness of the proposed scheme and
to compare its performance against those of the standard Joint and Dual EnKFs. The experimental





results suggest that the Dual-EnKF$_{\text{OSA}}$ is more robust, successfully estimating the hydraulic head and the conductivity field under different modeling scenarios. Sensitivity analyses demonstrate that

when more observations are assimilated, the Dual-EnKF$_{\text{OSA}}$ becomes more effective and significantly outperforms the standard Joint- and Dual-EnKF schemes. In addition, when using a sparse observation network in the aquifer domain, the accuracy of the Dual-EnKF$_{\text{OSA}}$ estimates is better preserved, unlike the Dual-EnKF, which seems to be more sensitive to the number of hydraulic wells. Moreover, the Dual-EnKF$_{\text{OSA}}$ results are shown to be more robust against observation noise.

On average, the Dual-EnKF$_{\text{OSA}}$ scheme leads to around 10% more accurate state and parameters solutions than those resulting from the standard Joint- and Dual-EnKFs.

The proposed scheme is easy to implement and only requires minimal modifications to a standard EnKF code. It is further computationally feasible, requiring only a marginal increase in the computational cost compared to the Dual-EnKF. This scheme should therefore be beneficial to the hydrology

community given its consistency, high accuracy, and robustness to changing modeling conditions. It could serve as an efficient estimation tool for real-world problems, such as groundwater, contaminant transport and reservoir monitoring, in which the available data are often sparse and noisy. Potential future research includes testing the Dual-EnKF$_{\text{OSA}}$ with realistic large-scale groundwater, contaminant transport and reservoir monitoring problems. Furthermore, combining the proposed

state-parameter estimation scheme with other iterative and hybrid ensemble approaches may be a promising direction for further improvements.

**Appendix A**

The following classical results of random sampling are extensively used in the derivation of the ensemble-based filtering algorithms presented in this paper.

**Property 1** (Hierarchical sampling (Robert, 2007)). Assuming that one can sample from $p(\mathbf{x}_1)$ and $p(\mathbf{x}_2|\mathbf{x}_1)$, then a sample, $\mathbf{x}_2^*$, from $p(\mathbf{x}_2)$ can be drawn as follows:

1. $\mathbf{x}_1^* \sim p(\mathbf{x}_1)$;

2. $\mathbf{x}_2^* \sim p(\mathbf{x}_2|\mathbf{x}_1^*)$.

**Property 2** (Conditional sampling (Hoffman and Ribak, 1991)). Consider a Gaussian pdf, $p(\mathbf{x}, \mathbf{y})$,

with $\mathbf{P}_{xy}$ and $\mathbf{P}_y$ denoting the cross-covariance of $\mathbf{x}$ and $\mathbf{y}$ and the covariance of $\mathbf{y}$, respectively. Then a sample, $\mathbf{x}^*$, from $p(\mathbf{x}|\mathbf{y})$, can be drawn as follows:

1. $(\widetilde{\mathbf{x}}, \widetilde{\mathbf{y}}) \sim p(\mathbf{x}, \mathbf{y})$;

2. $\mathbf{x}^* = \widetilde{\mathbf{x}} + \mathbf{P}_{xy}\mathbf{P}_y^{-1}[\mathbf{y} - \widetilde{\mathbf{y}}]$.





## Appendix B

We show here that the samples, $\widetilde{\mathbf{x}}_n^{(m)}(\mathbf{x}_{n-1}, \theta)$, given in (32), are drawn from the *a posteriori* transition pdf, $p(\mathbf{x}_n|\mathbf{x}_{n-1}, \theta, \mathbf{y}_n)$. Lets start by showing how Eqs. (30)-(31) are obtained. According to (15), on can show that the members, $\widetilde{\xi}_n^{(m)}(\mathbf{x}_{n-1}, \theta)$, given by (30), are samples from the transition pdf, $p(\mathbf{x}_n|\mathbf{x}_{n-1}, \theta) = \mathcal{N}_{\mathbf{x}_n}(\mathcal{M}_{n-1}(\mathbf{x}_{n-1}, \theta), \mathbf{Q}_{n-1})$. Furthermore, one may use Property 1 in (22), which is recalled here,

$$p(\mathbf{y}_n|\mathbf{x}_{n-1}, \theta) = \int \underbrace{p(\mathbf{y}_n|\mathbf{x}_n)}_{\mathcal{N}_{\mathbf{y}_n}(\mathbf{H}_n\mathbf{x}_n, \mathbf{R}_n)} \underbrace{p(\mathbf{x}_n|\mathbf{x}_{n-1}, \theta)}_{\approx \left\{\widetilde{\xi}_n^{(m)}(\mathbf{x}_{n-1}, \theta)\right\}_{m=1}^{N_e}} d\mathbf{x}_n, \tag{B1}$$

to obtain the members, $\widetilde{\mathbf{y}}_n^{(m)}(\mathbf{x}_{n-1}, \theta)$, given by (31); such members are, indeed, samples from $p(\mathbf{y}_n|\mathbf{x}_{n-1}, \theta)$.

Now, using the samples $\widetilde{\xi}_n^{(m)}(\mathbf{x}_{n-1}, \theta)$ of $p(\mathbf{x}_n|\mathbf{x}_{n-1}, \theta) = p(\mathbf{x}_n|\mathbf{x}_{n-1}, \theta, \mathbf{y}_{0:n-1})$ and the samples $\widetilde{\mathbf{y}}_n^{(m)}(\mathbf{x}_{n-1}, \theta)$ of $p(\mathbf{y}_n|\mathbf{x}_{n-1}, \theta) = p(\mathbf{y}_n|\mathbf{x}_{n-1}, \theta, \mathbf{y}_{0:n-1})$, one can apply Property 2 to the joint pdf,

$p(\mathbf{x}_n, \mathbf{y}_n|\mathbf{x}_{n-1}, \theta, \mathbf{y}_{0:n-1})$, assuming it Gaussian, to show that the samples $\widetilde{\mathbf{x}}_n^{(m)}(\mathbf{x}_{n-1}, \theta)$, given in (32), are drawn from the *a posteriori* transition pdf, $p(\mathbf{x}_n|\mathbf{x}_{n-1}, \theta, \mathbf{y}_{0:n}) = p(\mathbf{x}_n|\mathbf{x}_{n-1}, \theta, \mathbf{y}_n)$.

*Acknowledgements.* Research reported in this publication was supported by King Abdullah University of Science and Technology (KAUST).





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





**Table 1.** Approximate computational complexities of the Joint-EnKF, the Dual-EnKF, and the Dual-EnKF$_\mathrm{OSA}$ algorithms. Notations are as follows. $N_x$: number of state variables, $N_\theta$: number of parameter variables, $N_y$: number of observations, $N$: number of assimilation cycles, $N_e$: ensemble size, $\mathcal{C}_x$: state model cost ($= N_x^2$ is the linear KF), $\mathcal{C}_\theta$: parameter model cost (usually free $\equiv$ identity), $\mathcal{C}_y$: observation operator cost ($= N_y N_x$ in the linear KF), $\mathcal{S}_x$: storage volume for one state vector, $\mathcal{S}_\theta$: storage volume for one parameter vector.

| Algorithm | Time-update | Measurement-update | Storage |
|---|---|---|---|
| *Joint-EnKF* | $NN_e\left(\mathcal{C}_x + \mathcal{C}_\theta\right)$ | $NN_e\left(\mathcal{C}_y + N_y N_\theta\right) + NN_e^2\left(N_x + N_\theta\right)$ | $2NN_e\left(\mathcal{S}_x + \mathcal{S}_\theta\right)$ |
| *Dual-EnKF* | $NN_e\left(2\mathcal{C}_x + \mathcal{C}_\theta\right)$ | $2NN_e\mathcal{C}_y + NN_e^2\left(N_x + N_\theta\right)$ | $2NN_e\left(\mathcal{S}_x + \mathcal{S}_\theta\right)$ |
| *Dual-EnKF$_\mathrm{OSA}$* | $NN_e\left(2\mathcal{C}_x + \mathcal{C}_\theta\right)$ | $2NN_e\mathcal{C}_y + NN_e^2\left(2N_x + N_\theta\right)$ | $2NN_e\left(\mathcal{S}_x + \mathcal{S}_\theta\right)$ |

**Table 2.** Parameters of the random functions for modeling the spatial distributions of the reference and perturbed recharge fields. The ranges in $x$ and $y$ directions for the variorum model are given by $\lambda_x$ and $\lambda_y$, respectively. $\tau$ denotes the rotation angle of one clockwise rotation around the positive y-axis.

| Recharge | Mean | Variance | Variogram | $\lambda_x$ | $\lambda_y$ | $\tau$ |
|---|---|---|---|---|---|---|
| Reference Field | -20 (m³/day) | 1.03 (m³/day)² | Gaussian | 50 (m) | 100 (m) | 45° |
| Perturbed Field | -20 (m³/day) | 1.21 (m³/day)² | Gaussian | 50 (m) | 50 (m) | 45° |

**Table 3.** Mean average ensemble spread (AESP) of the water head and the hydraulic conductivity for three different ensemble sizes. The reported values are given for the Joint-EnKF, Dual-EnKF and the proposed Dual-EnKF$_\mathrm{OSA}$.

| | Hydraulic Head | | | Conductivity | | |
|---|---|---|---|---|---|---|
| | $N_e = 50$ | $N_e = 100$ | $N_e = 300$ | $N_e = 50$ | $N_e = 100$ | $N_e = 300$ |
| *Joint-EnKF* | 0.12294 | 0.144 | 0.20014 | 1.0763 | 1.0144 | 0.95128 |
| *Dual-EnKF* | 0.1256 | 0.14469 | 0.20081 | 1.0745 | 1.0155 | 0.95129 |
| *Dual-EnKF$_\mathrm{OSA}$* | 0.11737 | 0.14125 | 0.18259 | 1.0388 | 0.90654 | 0.8791 |





**Table 4.** Filter inbreeding indicator: Ratio of the mean average-absolute-error (AAE) and mean average-ensemble-spread (AESP) of the water head and the hydraulic conductivity for three different ensemble sizes. The reported values are given for the Joint-EnKF, Dual-EnKF and the proposed Dual-EnKF$_{\text{OSA}}$.

| | **Hydraulic Head** | | | **Conductivity** | | |
|---|---|---|---|---|---|---|
| | $N_e = 50$ | $N_e = 100$ | $N_e = 300$ | $N_e = 50$ | $N_e = 100$ | $N_e = 300$ |
| *Joint-EnKF* | 1.734 | 1.680 | 1.619 | 1.539 | 1.507 | 1.134 |
| *Dual-EnKF* | 1.449 | 1.443 | 1.360 | 1.123 | 1.123 | 0.834 |
| *Dual-EnKF*$_{\text{OSA}}$ | 0.805 | 0.802 | 0.854 | 0.793 | 0.792 | 0.801 |





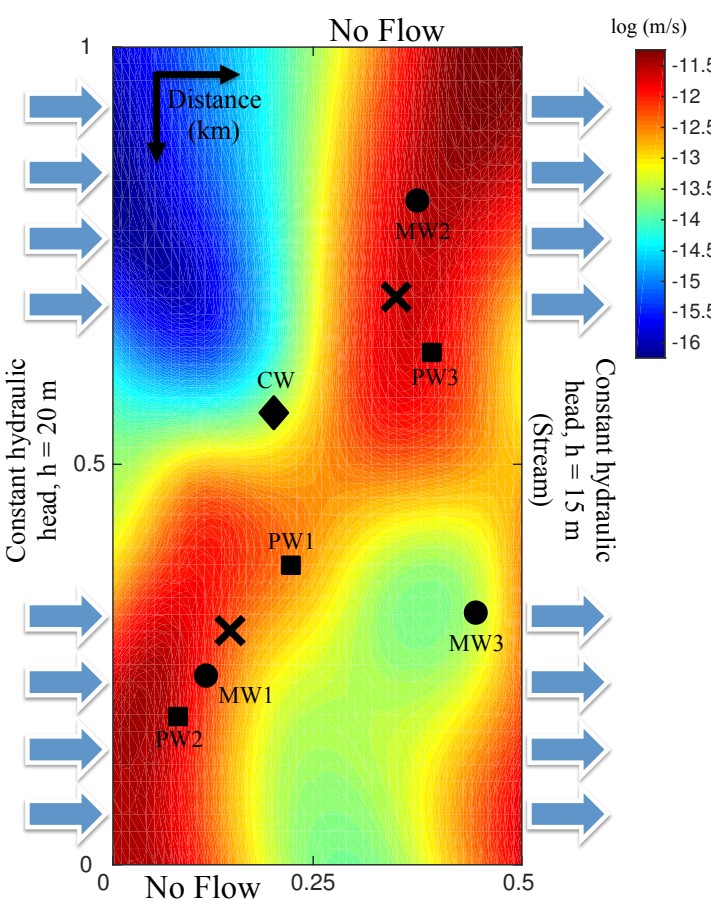

**Figure 1.** Plan view of the conceptual model for the 2D transient groundwater flow problem. East and west boundaries are Dirichlet with a given prescribed hydraulic heads. North and south boundaries are impermeable (no flow boundaries). The reference log-conductivity field obtained using the sequential Gaussian simulation code (Gómez-Hernández and Journel, 1993). A Gaussian variogram model is considered with a mean of -13 log(m/s), variance of 1.5 log(m$^2$/s$^2$), and range equal to 250 m and 500 m in the x and y directions, respectively. The black squares represent the pumping wells whereas the black circles denote the position of 3 monitoring wells. The black diamond is a control well.





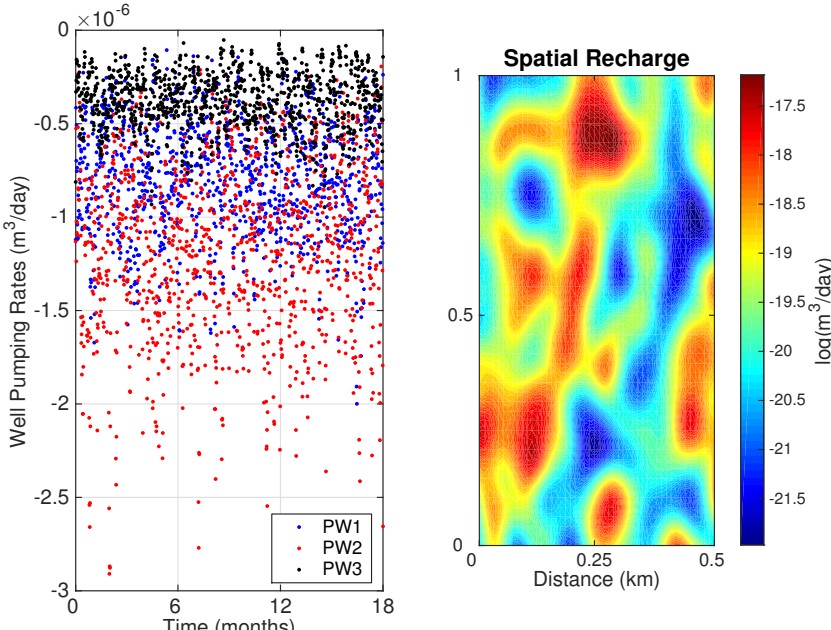

**Figure 2.** Left-Panel: Daily transient reference pumping rates from wells PW1, PW2 and PW3. Negative values indicate pumping or groundwater that is being removed from the aquifer. Right-Panel: Reference heterogenous spatial recharge values obtained using the sequential Gaussian simulation code (Gómez-Hernández and Journel, 1993) with parameters given in Table 2.




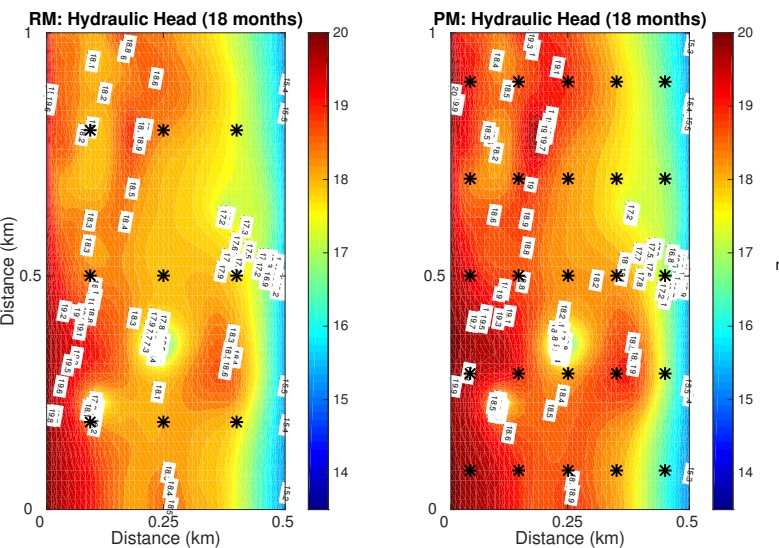

**Figure 3.** Groundwater flow contour maps obtained using the reference run (left panel) and the perturbed fore-cast model (right panel) after 18 months of simulation. The well locations from which head data are extracted are shown by black asterisks. In the left panel, we show the first network consisting of nine wells. In the right panel, the other network with 25 wells is displayed.



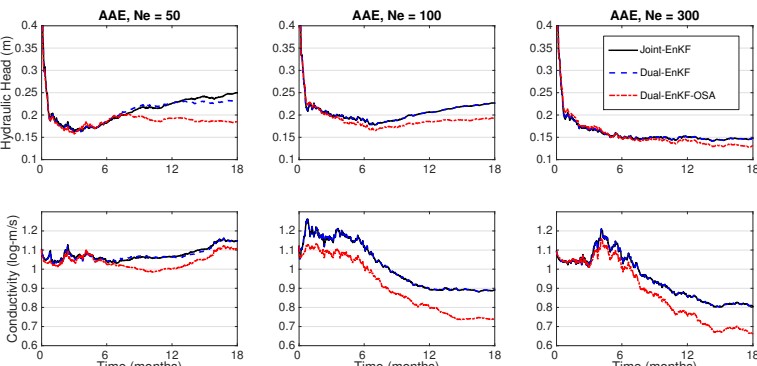

**Figure 4.** AAE time-series of the hydraulic head and conductivity using the Joint-EnKF, Dual-EnKF and Dual-EnKF$_{OSA}$. Results are shown for 3 scenarios in which assimilation of hydraulic head data are obtained from nine wells every 0.5 days. The three experimental scenarios use 50, 100 and 300 ensemble members with 0.50 m as the measurement error.





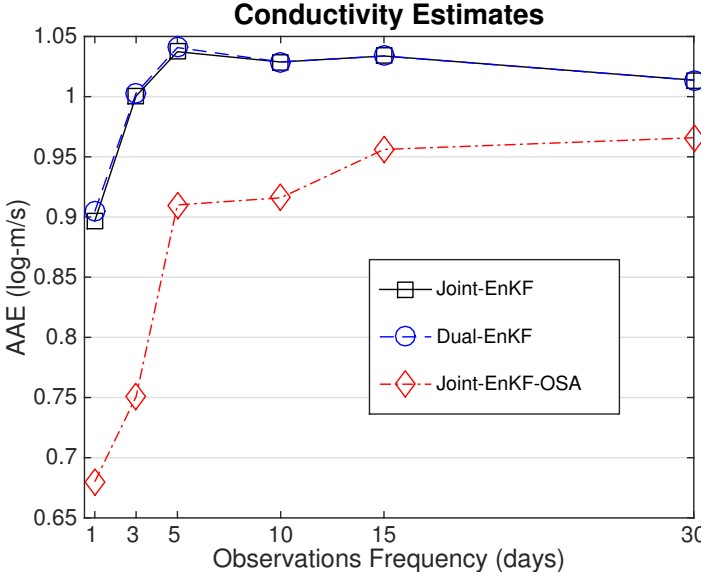

**Figure 5.** Mean average absolute errors (AAE) of log-hydraulic conductivity, $\log(K)$, obtained using the Joint-EnKF, Dual-EnKF, and Dual-EnKF$_{\mathrm{OSA}}$ schemes. Results are shown for 6 different scenarios in which assimilation of hydraulic head data are obtained from nine wells every 1, 3, 5, 10, 15 and 30 days. All 6 experimental scenarios use 100 ensemble members and 0.10 m as the measurement error.





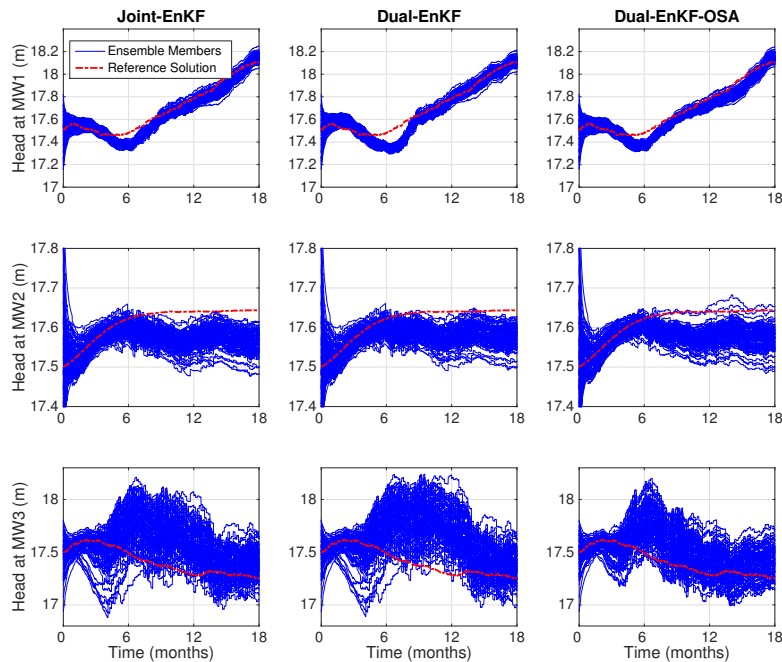

**Figure 6.** Reference (dashed) and predicted (solid) hydraulic head evolution at monitoring wells MW1, MW2, and MW3. Results are obtained using the Joint-EnKF and the Dual-EnKF$_{\text{OSA}}$ schemes with 100 members, 5 days as observation frequency, 9 observation wells, and 0.10 m of measurement noise.





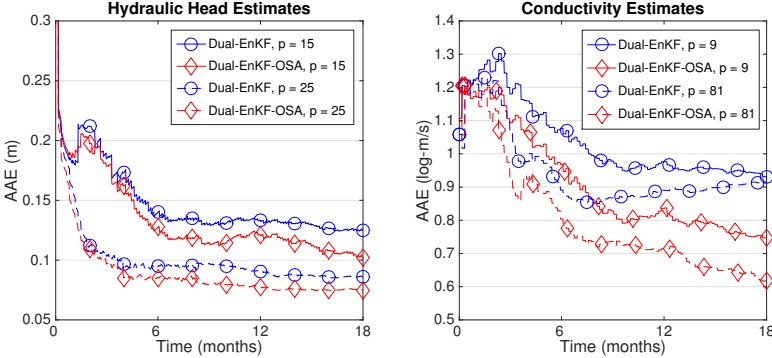

**Figure 7.** Time series of AAE of hydraulic head (left panel) and conductivity (right panel) using the Dual-EnKF and Dual-EnKF$_{\text{OSA}}$ schemes. Results are shown for 4 scenarios in which assimilation of hydraulic head data are obtained from 9, 15, 25 and 81 wells (uniformly distributed throughout the aquifer domain) every 5 days. The four experimental scenarios use 100 ensemble members and 0.10 m as the measurement error. The number of wells is denoted by $p$.





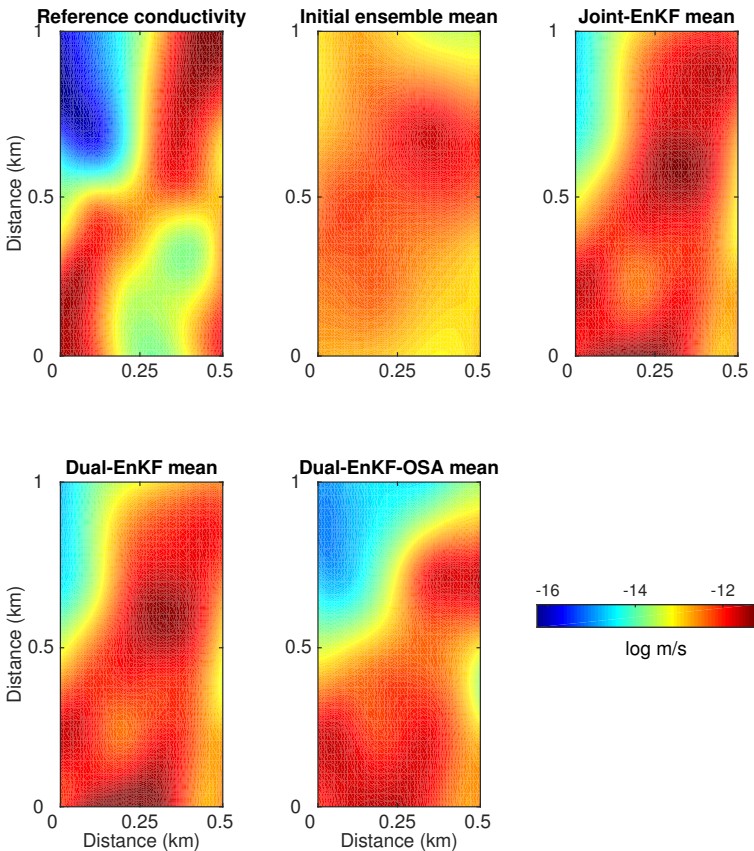

**Figure 8.** Spatial maps of the reference, initial and recovered ensemble means of hydraulic conductivity using the Joint-EnKF, Dual-EnKF, and Dual-EnKF$_{\text{OSA}}$ schemes. Results are shown for a scenario in which assimilation of hydraulic head data is obtained from nine wells every five days. This experiment uses 100 ensemble members and 0.10 m as the measurement error.





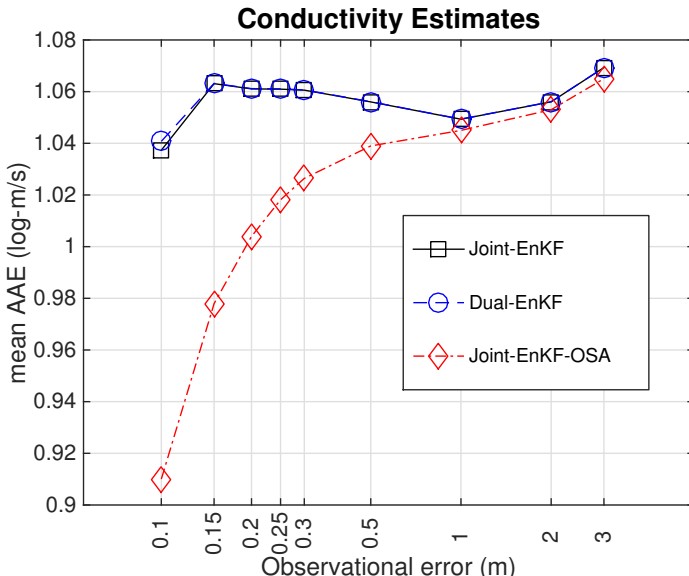

**Figure 9.** Mean AAE of the hydraulic conductivity using the Joint-EnKF, Dual-EnKF and Dual-EnKF$_{OSA}$ schemes. Results are shown for 9 different scenarios in which assimilation of hydraulic head data are obtained from 9 wells with measurement errors of 0.10, 0.15, 0.20, 0.25, 0.3, 0.5, 1, 2 and 3 m. The four experimental scenarios use 100 ensemble members and 5 days as observation frequency.





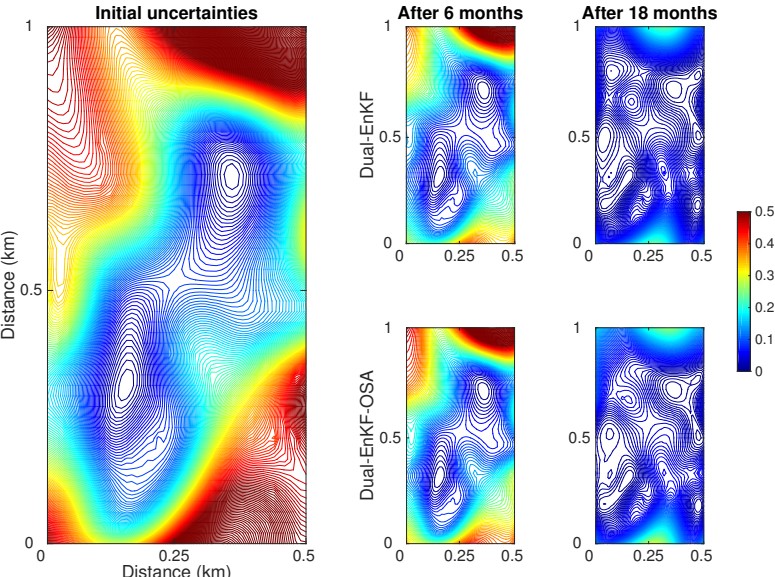

**Figure 10.** Left panel: Ensemble variance map of the initial conductivity field. Right sub-panels: Ensemble variance maps of estimated conductivity after 6 and 18 month assimilation periods using the Dual-EnKF and the proposed Dual-EnKF$_{OSA}$ schemes. These results are obtained with 100 members, 5 days of observation frequency, 9 observation wells, and 0.10 m as measurement noise.





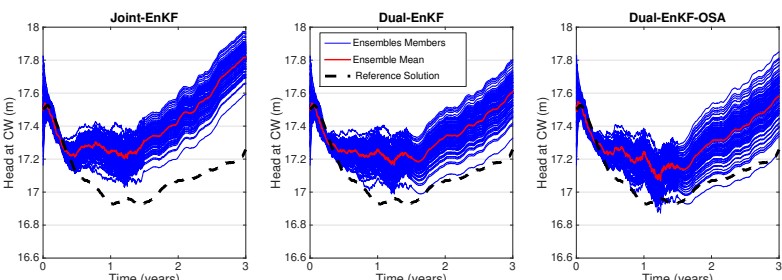

**Figure 11.** Reference (dashed) and predicted (solid) hydraulic head evolution at the control well: CW. Results are obtained using the Joint-EnKF, Dual-EnKF and the Dual-EnKF$_{OSA}$ schemes with 100 members, 1 day as observation frequency, 25 observation wells, and 0.50 m of measurement noise. The last 18 months are purely based on the forecast model prediction with no assimilation of data.





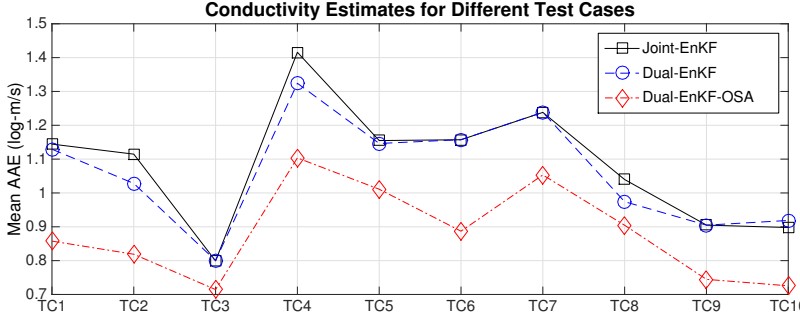

**Figure 12.** Performance of the Joint-/Dual-EnKF and the proposed Dual-EnKF$_{OSA}$ schemes in 10 different test cases (TC1, TC2, ...). Mean AAE of the conductivity estimates are displayed. These results are obtained with 100 members, 3 days of observation frequency, 9 observation wells, and 0.10 m as measurement noise.