# Peer review of "A Bayesian Consistent Dual Ensemble Kalman Filter for State-Parameter Estimation in Subsurface Hydrology"

_Hydrology and Earth System Sciences, 2015_

## Referee Comment (RC1) · Anonymous Referee #1 · 8 Mar 2016

General evaluation:

This paper is of potential interest to HESS. In general, the paper is well written and organized. The results support the proposed improved methodology. I have only one major concern. This is the difference with the paper of Gharamti et al., 2015 in Journal of Hydrology. I understand that the methodology is already introduced there, and that now the mathematical-statistical basis of it is improved. In addition a new rigorous synthetic study was carried out. The authors should exactly point out what is new in this paper and motivate why this warrants a new publication. If answered satisfactory, the paper can be published with minor revisions.

Detailed comments:

[Figure]

L40: "have been proposed" instead of "has been proposed".

L64: "was given" instead of "was carried"

L99-L101: Rephrase.

L109-L110: Change to: "(. . .) various experiment settings and observation scenarios."

L122: this should be t(n-1) to t(n)?

L203-L206: This was not found in Song et al. (2015, VZJ). There Dual EnKF performed worse, and only a rigorous Restart EnKF gave better results. Reformulate.

L377-L378: The pumping rate is unfortunately unrealistic low. It would have been better if the authors would have worked with a more realistic case.

L587-L592: I do not see many differences and these are probably related to the initial conditions after the assimilation phase. Reconsider this text part.

Caption Figure 1. Change to: "The reference log-conductivity field was obtained (. . .)"

Caption Figure 9. Why does AAE not decrease for joint EnKF and dual EnKF for small observation errors? Please comment.

Caption Figure 9: "are obtained" is not correct.

Caption Figure 10. Why do you use lines in the figures? The legend is not clear.

———————————————————

---

## Referee Comment (RC2) · Anonymous Referee #2 · 14 Mar 2016

A one-step-ahead smoothing based dual EnKF is presented for state-parameter estimation of groundwater flow models. The performance of the algorithm is tested through a set of sensitivity analysis and comparison with standard joint and dual EnKF. The paper is well written and I have only a few questions.

Main concerns

1. A one-step-ahead smoothing based dual EnKF is presented in the manuscript. The authors compared results of the new method with standard joint and dual EnKF. As mentioned by the authors, Gharamti et al (2015) proposed a new EnKF method too by combining the one-step-ahead smoothing formulation (page 3). What will the result look like if these two one-step-ahead based EnKF methods are compared? Also the

dual EnKF OSA needs to be better distinguished from the one by Gharamti et al (2015).

2. Page 8. The authors talked about one-step-ahead smoothing function but did not explain what is this function used for? What role does it play in the new algorithm dual EnKF OSA and how does it work?

3. Page 11. The observation data are used three time in dual EnKF OSA rather than twice as in the dual EnKF. The authors thus concluded that it is in a fully consistent Bayesian formulation. Please clarify the relationship between them. Also please relate this with the comment on standard dual EnKF that the ensemble does not represent the forecast pdf (page 8 lines 231-233).

4. Page 11 line 527-528 "The proposed dual EnKF OSA efficiently recovers the reference trajectory of MW2 and MW3". I think this statement is not so proper since the trajectory or trend of the reference is not captured well by any method, including the dual EnKF OSA. The reference is barely covered by the ensemble and the peak values are late in the ensemble at MW3. But I agree that the dual EnKF OSA works better than the other two methods.

5. Page 17 line 534 "We further examine . . ...against the joint- and dual- EnKFs . . .." But in Figure 7 only the results by dual-EnKF and dual- EnKF OSA are shown. The results by joint EnKF are not included.

6. Page 18 line 576 " the dual- EnKF OSA tends to maintain a larger variance at the edges than the dual-EnKF, which in turn increases the impact of the observations". I have two questions on this: first, it is difficult to tell (Figure 10) the "larger" variance by dual- EnKF OSA. Their results look pretty similar. Secondly, why will the variance of hydraulic conductivity at the edge increase the impact of the observations (at 9 wells) which is not so near from the edges? Furthermore, the boundary conditions are either no-flow or constant head which have limited influence on observations. One suggestions on the color bar of the figure: the white color does not appear in the bar but occupy a lot of area in the Figure and the contours make the figure complicated.

Minor corrections

1. Page 2 line 38, "Hendricks Franssen and Kinzelbach, 2009" instead of "Franssen and Kinzelbach, 2009". At the same time, please correct the item in the reference list (page 22).

2. Page 5 line 150. "Let, for an ensemble . . ., r denotes" should be "denote" instead of "denotes"?

3. Page 10. The equation 25 looks exactly the same as equation 5. Is this correct?

4. Figure 1, the black crosses represent hard measurements according to the text on Line 425. But it can also be added here to avoid any confusion.

5. Title of the subsection 5.5 "further assessment of the dual- EnKF OSA scheme" does not reflect the content. From the title we expect the result by dual- EnKF OSA only. But in fact it still compares the results of three methods. It could be changed to "prediction capability assessment" or something like this.
* * *

---

## Author Comment (AC1) · 30 Mar 2016

- **Reply to Referee #1** -

We would like to thank the referee for his/her valuable and constructive comments. These greatly improved the quality of our manuscript and helped us clarify several points, which we gratefully acknowledge. All the referee's comments were considered in the revised manuscript and detailed replies are given below.

[Figure]

**General evaluation**

C. *This paper is of potential interest to HESS. In general, the paper is well written and organised. The results support the proposed improved methodology. I have only one major concern. This is the difference with the paper of Gharamti et al., 2015 in Journal of Hydrology. I understand that the methodology is already introduced there, and that now the mathematical-statistical basis of it is improved. In addition a new rigorous synthetic study was carried out. The authors should exactly point out what is new in this paper and motivate why this warrants a new publication. If answered satisfactory, the paper can be published with minor revisions.*

R. We thank the referee for acknowledging our work and for the comment he/she raised. The proposed Dual-EnKF$_{OSA}$ in this work results from the generic algorithm of Section 3.1 by applying two random sampling properties (see Appendix A) under the Gaussian assumption. At each assimilation step of the Dual-EnKF$_{OSA}$, the observed data are used three times through Kalman-like updates: twice in the smoothing step (one for smoothing the previous state as in Eq. (27) and one for updating the parameters as in Eq. (28)), and once in the "forecast" step to compute the analysis of the current state as in Eq. (35).

The work in *Gharamti et al. (2015)* follows a similar approach and applies the same two random sampling properties on the generic algorithm of Section 3.1, but under the following assumption (beside the Gaussian assumption)[1]:

$$p(x_n|x_{n-1},\theta,y_n) = p(x_n|x_{n-1},\theta), \tag{1}$$

which is based on the fact that given the previous state, $x_{n-1}$, and the parameters, $\theta$, the current state, $x_n$, is independent of its observation, $y_n$. Following this assumption,
* * *
[1]Refer to Eq. (16) in *Gharamti et al. (2015)*.

the "forecast" step of the generic algorithm which is composed of a Bayesian step (24) followed by a propagation step (23), reduces to the propagation step (23). As stated in page 445 of *Gharamti et al. (2015)*, the assumption (1) has been adopted to compute the analysis pdf, $p(x_n|y_{0:n})$, from the joint smoothing pdf, $p(x_{n-1}, \theta|y_{0:n})$, based on Eq. (9) (or Eq. (23) in our manuscript), by avoiding the use of the computationally demanding term $p(x_n|x_{n-1}, \theta, y_n)$ and replace it by the more easily sampled state transition pdf, $p(x_n|x_{n-1}, \theta)$. Here, we propose a more efficient approach (see pages 10-11 and Appendix B) to directly sample from the analysis pdf without explicitly computing $p(x_n|x_{n-1}, \theta, y_n)$ and without the need of any additional assumption.

Now, when applying the two random sampling properties (under the Gaussian assumption), the Joint-EnKF$_{OSA}$ in *Gharamti et al. (2015)* shares the same smoothing step as the Dual-EnKF$_{OSA}$, which, as mentioned above, involves two Kalman-like updates. However, in contrast with the Dual-EnKF$_{OSA}$, the Joint-EnKF$_{OSA}$ does not involve a Kalman-like update in the "forecast" step (because of the omission of the Bayesian step (24) from the generic algorithm). In terms of computational cost, these algorithms have almost the same cost as they require the same number of model runs; the only difference is one Kalman update for each member which is generally computationally not consequent compared to the cost of integrating the model. This further allows to explicitly put in context the conditions under which the (heuristic) steps of the standard Dual-EnKF can be derived in a Bayesian setting.

To summarise, the proposed Dual-EnKF$_{OSA}$ is more general than the Joint-EnKF$_{OSA}$ of *Gharamti et al. (2015)*, inasmuch as it involves one more Kalman-like update. This was made clearer in lines 332-343 of the revised manuscript. Moreover and as a way of illustrating the difference between the two schemes, we have included additional experiments results using the Joint-EnKF$_{OSA}$. We showed that the proposed Dual-EnKF$_{OSA}$ slightly outperforms the Joint-EnKF$_{OSA}$ ($\sim 5\%$ better accuracy). We further reported the average ensemble spread, as estimated by the Joint-EnKF$_{OSA}$, for both states and parameters in Table 1.

**Detailed comments**

C1.  *L40: "have been proposed" instead of "has been proposed".*

R1.  Done. Thank you.

C2.  *L64: "was given" instead of "was carried"*

R2.  Done. Thank you.

C3.  *L99-L101: Rephrase.*

R3. We have rephrased the sentence which now reads as follows: "Our goal is to derive a new Dual-EnKF-like algorithm that retains the separate formulation of the state and parameters update steps, within a fully Bayesian framework."

C4.  *L109-L110: Change to: "(...) various experiment settings and observation scenarios."*

R4.  Done. Thank you.

C5.  *L122: this should be t(n-1) to t(n)?*

R5.  Based on our notation in system (1), our statement in L122 is correct: "$\mathcal{M}_n$ *is a*

*nonlinear operator integrating the system state from time $t_n$ to $t_{n+1}$".*

C6. *L203-L206: This was not found in Song et al. (2015, VZJ). There Dual EnKF performed worse, and only a rigorous Restart EnKF gave better results. Reformulate.*

R6. We thank the referee for pointing this out. In fact, we have read the article by *Song et al. (2015)* and we were aware of this study prior to our submission. The authors tested the use of the Confirming Step EnKF, the restart EnKF and a modified variant of the restart EnKF. Unlike the Joint and the Dual EnKFs, these filters only update the parameters using a Kalman-type analysis. The state ensemble members, on the other hand, are obtained after integrating the model. Comparing the Confirming-Step-EnKF of *Wen and Chen (2006)*, which we cite in the manuscript, to that of *Song et al. (2015)* may not be that straightforward considering the differences between their models. On the one hand, *Wen and Chen (2006)* worked with a reservoir simulator and *Song et al. (2015)* used an unsaturated flow problem. It is possible that the efficiency of the Confirming-Step becomes more pronounced in nonlinear reservoir systems and strongly heterogenous subsurface formations, which is not the case in *Song et al. (2015)*. To avoid the confusion, we have removed the confirming-step EnKF reference from the discussion and reformulated the sentence.

C7. *L377-L378: The pumping rate is unfortunately unrealistic low. It would have been better if the authors would have worked with a more realistic case.*

R7. We fully understand the concern of the referee. The choice of the well pumping rates was based on contributions from the initial head values in addition to the present recharge. Starting from a uniform initial hydraulic head $h_0 = 15$ m, the recharge and the pumping rates eventually yield heterogeneous spatial distribution of $h$, varying between

$\sim 13$ to $20$ m, as shown in Figure 3. Increasing the pumping rates in our setup may, however, cause clear suction of the groundwater (negative head) and the dynamics will be mainly dominated by this forcing term. The considered rates create enough variability in space and time to test the assimilations schemes. Thank you.

C8. *L587-L592: I do not see many differences and these are probably related to the initial conditions after the assimilation phase. Reconsider this text part.*

R8. We are not sure what the referee exactly mean by the initial conditions at the end of assimilation. In fact, after the assimilation phase we simply continue the simulation in a prediction mode without assimilating any data. Thus, we don't tamper or impose any changes on the hydraulic head obtained after assimilation. In terms of the differences, we disagree with the referee because the Joint-EnKF shows a clear overestimation of $h$ at the control well unlike the Dual-EnKF and more importantly the proposed Dual-EnKF$_{OSA}$. To illustrate, we evaluated the absolute bias during the prediction phase for the three schemes. We plot the resulting curves in Figure 1 below. As shown, the bias suggested by the proposed scheme is the smallest and around 0.5 m less than that of the Joint-EnKF. To support our argument, we have included this figure in the manuscript and we interpreted the results accordingly. Thank you.

C9. *Caption Figure 1. Change to: "The reference log-conductivity field was obtained (...)"*

R9. Done. Thank you.

C10. *Caption Figure 9. Why does AAE not decrease for joint EnKF and dual EnKF*

*for small observation errors? Please comment.*

R10.   The referee is raising an interesting point here. The performance of the Joint-EnKF and the Dual-EnKF clearly degrades when the observation error decreases, which might seem as counterintuitive. However, the errors only increase from 1.05 to 1.06 which is not significant. When the observational error decreases further to 0.1 m, the AAE of conductivity decreases to approximately 1.04. To test this further, we examined the performance of the filters with even smaller observation error, i.e., $0.05$ m. We updated the figure in the revised manuscript, accordingly. We notice that the performance of the Joint-EnKF and the Dual-EnKF continue to improve reaching an AAE of around 1.01. Regarding the performance for measurement errors between 0.1 and 0.15, this could possibly be due to statistical fluctuations related to this particular case and setup.

C11.   *Caption Figure 9: "are obtained" is not correct.*

R11.   Rephrased. Thank you.

C12.    *Caption Figure 10. Why do you use lines in the figures? The legend is not clear.*

R12.   The figure and its legend have been updated following the referee's comment.

Please also note the supplement to this comment:
http://www.hydrol-earth-syst-sci-discuss.net/hess-2015-544/hess-2015-544-AC1-supplement.zip

[Figure]

[Figure]

**Absolute Bias: Hydraulic Head**

(Figure: plot of Control Well (m) versus Time (years) showing Joint-EnKF, Dual-EnKF, and Dual-EnKF-OSA curves)

**Fig. 1.** Absolute bias resulting from the three assimilation schemes during the prediction phase at the control well.

---

## Author Comment (AC2) · 30 Mar 2016

**- **Reply to Referee #2** -**

We would like to thank the referee for carefully reviewing our work and for his/her constructive comments. We have revised the manuscript taking into account all the referee comments and suggestions whose detailed replies are given below.

**Main concerns**

C1.  *A one-step-ahead smoothing based dual EnKF is presented in the manuscript. The authors compared results of the new method with standard joint and dual EnKF. As mentioned by the authors, Gharamti et al (2015) proposed a new EnKF method too by combining the one-step-ahead smoothing formulation (page 3). What will the result look like if these two one-step-ahead based EnKF methods are compared? Also the dual EnKF OSA needs to be better distinguished from the one by Gharamti et al (2015).*

R1.  We thank the referee for bringing this up. The referee raised two points:

- *Comparison of the filtering schemes:*

  We have examined the performance of the Joint-EnKF$_{OSA}$ of *Gharamti et al. (2015)* compared to the new filtering schemes in the first section of the results. We plot the resulting state and parameter estimates on top of the previous ones in Figure 4 and Table 3. The Joint-EnKF$_{OSA}$ is found more reliable during the early assimilation period but later and towards the end of assimilation, the proposed new scheme becomes clearly more accurate.

- *Differences between the algorithms:*

  The proposed Dual-EnKF$_{OSA}$ in this work results from the generic algorithm of Section 3.1 by applying two random sampling properties (see Appendix A) under the Gaussian assumption. At each assimilation step of the Dual-EnKF$_{OSA}$, the observed data are used three times through Kalman-like updates: twice in the smoothing step (one for smoothing the previous state as in Eq. (27) and one for updating the parameters as in Eq. (28)), and once in the "forecast" step to compute the analysis of the current state as in Eq. (35).

The work in *Gharamti et al. (2015)* follows a similar approach and applies the same two random sampling properties on the generic algorithm of Section 3.1, but under the following assumption (beside the Gaussian assumption)[1]:

$$p(x_n|x_{n-1}, \theta, y_n) = p(x_n|x_{n-1}, \theta), \qquad (1)$$

which is based on the fact that given the previous state, $x_{n-1}$, and the parameters, $\theta$, the current state, $x_n$, is independent of its observation, $y_n$. Following this assumption, the "forecast" step of the generic algorithm which is composed of a Bayesian step (24) followed by a propagation step (23), reduces to the propagation step (23). As stated in page 445 of *Gharamti et al. (2015)*, the assumption (2) has been adopted to compute the analysis pdf, $p(x_n|y_{0:n})$, from the joint smoothing pdf, $p(x_{n-1}, \theta|y_{0:n})$, based on Eq. (9) (or Eq. (23) in our manuscript), by avoiding the use of the computationally demanding term $p(x_n|x_{n-1}, \theta, y_n)$ and replace it by the more easily sampled state transition pdf, $p(x_n|x_{n-1}, \theta)$. Here, we propose a more efficient approach (see pages 10-11 and Appendix B) to directly sample from the analysis pdf without explicitly computing $p(x_n|x_{n-1}, \theta, y_n)$ and without the need of any additional assumption.

Now, when applying the two random sampling properties (under the Gaussian assumption), the Joint-EnKF$_{OSA}$ in *Gharamti et al. (2015)* shares the same smoothing step as the Dual-EnKF$_{OSA}$, which, as mentioned above, involves two Kalman-like updates. However, in contrast with the Dual-EnKF$_{OSA}$, the Joint-EnKF$_{OSA}$ does not involve a Kalman-like update in the "forecast" step (because of the omission of the Bayesian step (24) from the generic algorithm). In terms of computational cost, these algorithms have almost the same cost as they require the same number of model runs; the only difference is one Kalman update for each member which is generally computationally not consequent compared to the cost of integrating the model. The differences between the proposed Dual-
* * *
[1]Refer to Eq. (16) in *Gharamti et al. (2015)*.

EnKF$_{OSA}$ and the Joint-EnKF$_{OSA}$ of *Gharamti et al. (2015)* were made clearer in lines 332-343 of the revised manuscript.

C2. *Page 8. The authors talked about one-step-ahead smoothing function but did not explain what is this function used for? What role does it play in the new algorithm dual EnKF OSA and how does it work?*

R2. We understand that the referee is referring to our statement (page 8, lines 245-246): "...*but also involves a (new) smoothing step that constraints the state with the future observation.*". This suggests that compared to the standard Dual-EnKF, the proposed Dual-EnKF$_{OSA}$ involves a new update step of the state using the future observation (hence the term "one-step-ahead (OSA) smoothing"). This smoothing "function" is given by Eq. (27), being a Kalman-like update of the state analysis members using the future observation. Line 317 of page 11 was updated to clarify the Kalman-like update character of the smoothing "function". Thank you.

C3. *Page 11. The observation data are used three time in dual EnKF OSA rather than twice as in the dual EnKF. The authors thus concluded that it is in a fully consistent Bayesian formulation. Please clarify the relationship between them. Also please relate this with the comment on standard dual EnKF that the ensemble does not represent the forecast pdf (page 8 lines 231-233).*

R3. We thank the referee for pointing this out. The proposed Dual-EnKF$_{OSA}$ is not a fully Bayesian algorithm because the observed data are used three times, but because this algorithm results from the generic (theoretically sound) Bayesian filtering algorithm presented in Section 3.1 by applying two random sampling properties (see Appendix

A), and under the (commonly used) Gaussian assumption only. As stated in our reply R1 above, at each assimilation time step of the resulting Dual-EnKF$_{OSA}$, it turns out that the observed data are used three times through Kalman-like updates: one for smoothing the previous state (Eq. (27)), one for updating the parameters (Eq. (28)) and one for updating the current state (Eq. (35)). Accordingly, the use of the observed data three times is a consequence of the fully Bayesian character of the proposed scheme and not a cause. This was made clearer in the text below Eq. (35).

Regarding our statement on standard Dual-EnKF that the ensemble does not represent the forecast pdf and therefore it is a heuristic algorithm (page 8, lines 231-233), please note that this is related to the (Bayesian consistent) Joint-EnKF and not to the Dual-EnKF$_{OSA}$. Indeed, as one can see from Sections 2.2.1 and 2.2.2, the Joint-EnKF and the Dual-EnKF algorithms mainly differ in the state analysis step. The analysis members of the state, $x_n^{a,(m)}$, are computed by the Joint-EnKF using a Kalman-like update of the forecast members, $x_n^{f,(m)} = \mathcal{M}_{n-1}(x_{n-1}^{a,(m)}, \theta_{|n-1}^{(m)})$ given in Eq. (5), while the Dual-EnKF computes $x_n^{a,(m)}$ following the same mechanism but using the members $\widetilde{x}_n^{f,(m)} = \mathcal{M}_{n-1}(x_{n-1}^{a,(m)}, \theta_{|n}^{(m)})$ given in Eq. (12), instead of $x_n^{f,(m)}$. However, in contrast with $x_n^{f,(m)}$, the members $\widetilde{x}_n^{f,(m)}$ are not samples from the forecast pdf (more generally, one does not know from which pdf $\widetilde{x}_n^{f,(m)}$ are sampled) since they are obtained by integrating the model with the updated parameters, $\theta_{|n}^{(m)}$ (along with $x_{n-1}^{a,(m)}$), instead of $\theta_{|n-1}^{(m)}$. Despite that, these are used in the analysis step as forecast members to compute the analysis members, $x_n^{a,(m)}$, following the same Kalman-like update as in the Joint-EnKF, hence the heuristic nature of the Dual-EnKF.

Now, to relate that with the Dual-EnKF$_{OSA}$, one can easily see that this latter reduces to the Dual-EnKF in the particular case of a perfect model and in the absence of state smoothing (i.e., when Eq. (27) vanishes and $x_{n-1}^{s,(m)} = x_{n-1}^{a,(m)}$ in Eq. (33)). This was made clearer in page 11, lines 330-331 of the revised manuscript.
C4.    *Page 17 line 527-528 "The proposed dual EnKF OSA efficiently recovers the reference trajectory of MW2 and MW3". I think this statement is not so proper since the trajectory or trend of the reference is not captured well by any method, including the dual EnKF OSA. The reference is barely covered by the ensemble and the peak values are late in the ensemble at MW3. But I agree that the dual EnKF OSA works better than the other two methods.*

R4.    We have relaxed our argument here and rephrased the sentence accordingly. The sentence now reads as follows: "The proposed Dual-EnKF$_{OSA}$ performs fairly well, providing a reasonable recovery of the reference trajectory at MW2 and MW3.". Thank you.

C5.    *Page 17 line 534 "We further examine ..... against the joint- and dual- EnKFs ...." But in Figure 7 only the results by dual-EnKF and dual- EnKF OSA are shown. The results by joint EnKF are not included.*

R5.    We thank the referee for pointing this out. We have now included the results from the Joint-EnKF in Figure 7.

C6.    *Page 18 line 576 "the dual- EnKF OSA tends to maintain a larger variance at the edges than the dual-EnKF, which in turn increases the impact of the observations". I have two questions on this: first, it is difficult to tell (Figure 10) the "larger" variance by dual- EnKF OSA. Their results look pretty similar. Secondly, why will the variance of hydraulic conductivity at the edge increase the impact of the observations (at 9 wells) which is not so near from the edges? Furthermore, the boundary conditions are either no-flow or constant head which have limited influence on observations. One suggestions on the color bar of the figure: the white color does not appear in the bar*

*but occupy a lot of area in the Figure and the contours make the figure complicated.*

R6. We have improved the quality of the figure. Essentially, we removed the contour lines and kept the colouring. We further adjusted the colour bar to better emphasize the different behaviours of the schemes. In terms of the larger variance at the north boundary, we refer to the fact that an ensemble with larger spread (variance) is expected to fit more the observations using a Kalman-based update. However, if the ensemble spread is very small and the estimate is still far from the truth then the impact of assimilating future data would be minimal.

**Minor corrections**

C1. *Page 2 line 38, "Hendricks Franssen and Kinzelbach, 2009" instead of "Franssen and Kinzelbach, 2009". At the same time, please correct the item in the reference list (page 22).*

R1. Done. Thank you.

C2. *Page 5 line 150. "Let, for an ensemble ..., $r$ denotes" should be "denote" instead of "denotes"?*

R2. Done, thank you.

C3. *Page 10. The equation 25 looks exactly the same as equation 5. Is this correct?*

R3. Yes, this is correct. However, mechanisms (5) and (25) are the same, but not their "outputs". In other words, the forecast members $x_n^{f,(m)}$ resulting from (5) are not the same as those obtained by (25) even when starting from the same "input" $(x_{n-1}^{a,(m)}, \theta_{|n-1}^{(m)})$ (which is very unlikely). This is because the noise $\eta_{n-1}^{(m)}$ in (5) are different than those of (25) even though they are sampled from the same Gaussian distribution $\mathcal{N}(0, Q_{n-1})$.

C4. *Figure 1, the black crosses represent hard measurements according to the text on Line 425. But it can also be added here to avoid any confusion.*

R4. Done. Thank you.

C5. *Title of the subsection 5.5 "further assessment of the dual- EnKF OSA scheme" does not reflect the content. From the title we expect the result by dual- EnKF OSA only. But in fact it still compares the results of three methods. It could be changed to "prediction capability assessment" or something like this.*

R5. We thank the referee for the comment. This title was changed as suggested.

Please also note the supplement to this comment:
http://www.hydrol-earth-syst-sci-discuss.net/hess-2015-544/hess-2015-544-AC2-supplement.zip

---

## Author Response (AR1)

Hydrology and Earth System Sciences (HESS) paper number hess-2015-544
**A Bayesian Consistent Dual Ensemble Kalman Filter for State-Parameter Estimation in Subsurface Hydrology**
by B. Ait-El-Fquih, M.E. Gharamti and I. Hoteit

Dear Professor Mauro Giudici,

We would like to thank you, as well as the referees, for all the constructive suggestions and comments you have made throughout the review process of our manuscript. We have done our best to address all referees' concerns, which greatly helped us improving the quality of our work. In particular, we have carefully revised the manuscript to emphasize the novelty of this new work with respect to our previous published papers. We have further described in details the differences with our work in Gharamti et al. (2015) (Journal of Hydrology). Numerical results comparing the new proposed scheme with that of Gharamti et al. (2015) have been also included in Figure 4 and Table 3, as suggested by Referee 2.

We thank you again for all your editorial efforts.

Sincerely yours,

B. Ait-El-Fquih, M.E. Gharamti and I. Hoteit

Hydrology and Earth System Sciences (HESS) paper number hess-2015-544
**A Bayesian Consistent Dual Ensemble Kalman Filter for State-Parameter Estimation in Subsurface Hydrology**
by B. Ait-El-Fquih, M.E. Gharamti and I. Hoteit

**- Reply to Referee #1 -**

We would like to thank the referee for his/her valuable and constructive comments. These greatly improved the quality of our manuscript and helped us clarify several points, which we gratefully acknowledge. All the referee's comments were considered in the revised manuscript and detailed replies are given below.

**General evaluation**

C. *This paper is of potential interest to HESS. In general, the paper is well written and organised. The results support the proposed improved methodology. I have only one major concern. This is the difference with the paper of Gharamti et al., 2015 in Journal of Hydrology. I understand that the methodology is already introduced there, and that now the mathematical-statistical basis of it is improved. In addition a new rigorous synthetic study was carried out. The authors should exactly point out what is new in this paper and motivate why this warrants a new publication. If answered satisfactory, the paper can be published with minor revisions.*

R. We thank the referee for acknowledging our work and for the comment he/she raised. The proposed Dual-EnKF$_{\text{OSA}}$ in this work results from the generic algorithm of Section 3.1 by applying two random sampling properties (see Appendix A) under the Gaussian assumption. At each assimilation step of the Dual-EnKF$_{\text{OSA}}$, the observed data are used three times through Kalman-like updates: twice in the smoothing step (one for smoothing the previous state as in Eq. (27) and one for updating the parameters as in Eq. (28)), and once in the "forecast" step to compute the analysis of the current state as in Eq. (35).
The work in *Gharamti et al. (2015)* follows a similar approach and applies the same two random sampling properties on the generic algorithm of Section 3.1, but under the following assumption (beside the Gaussian assumption)[1]:

$$p(\mathbf{x}_n|\mathbf{x}_{n-1}, \theta, \mathbf{y}_n) = p(\mathbf{x}_n|\mathbf{x}_{n-1}, \theta), \tag{1}$$

which is based on the fact that given the previous state, $\mathbf{x}_{n-1}$, and the parameters, $\theta$, the current state, $\mathbf{x}_n$, is independent of its observation, $\mathbf{y}_n$. Following this assumption, the "forecast" step of the generic algorithm which is composed of a Bayesian step (24) followed
* * *
[1]Refer to Eq. (16) in *Gharamti et al. (2015)*.

by a propagation step (23), reduces to the propagation step (23). As stated in page 445 of *Gharamti et al. (2015)*, the assumption (1) has been adopted to compute the analysis pdf, $p(\mathbf{x}_n|\mathbf{y}_{0:n})$, from the joint smoothing pdf, $p(\mathbf{x}_{n-1}, \theta|\mathbf{y}_{0:n})$, based on Eq. (9) (or Eq. (23) in our manuscript), by avoiding the use of the computationally demanding term $p(\mathbf{x}_n|\mathbf{x}_{n-1}, \theta, \mathbf{y}_n)$ and replace it by the more easily sampled state transition pdf, $p(\mathbf{x}_n|\mathbf{x}_{n-1}, \theta)$. Here, we propose a more efficient approach (see pages 10-11 and Appendix B) to directly sample from the analysis pdf without explicitly computing $p(\mathbf{x}_n|\mathbf{x}_{n-1}, \theta, \mathbf{y}_n)$ and without the need of any additional assumption.

Now, when applying the two random sampling properties (under the Gaussian assumption), the Joint-EnKF$_{\text{OSA}}$ in *Gharamti et al. (2015)* shares the same smoothing step as the Dual-EnKF$_{\text{OSA}}$, which, as mentioned above, involves two Kalman-like updates. However, in contrast with the Dual-EnKF$_{\text{OSA}}$, the Joint-EnKF$_{\text{OSA}}$ does not involve a Kalman-like update in the "forecast" step (because of the omission of the Bayesian step (24) from the generic algorithm). In terms of computational cost, these algorithms have almost the same cost as they require the same number of model runs; the only difference is one Kalman update for each member which is generally computationally not consequent compared to the cost of integrating the model. This further allows to explicitly put in context the conditions under which the (heuristic) steps of the standard Dual-EnKF can be derived in a Bayesian setting. To summarise, the proposed Dual-EnKF$_{\text{OSA}}$ is more general than the Joint-EnKF$_{\text{OSA}}$ of *Gharamti et al. (2015)*, inasmuch as it involves one more Kalman-like update. This was made clearer in lines 332-343 of the revised manuscript. Moreover and as a way of illustrating the difference between the two schemes, we have included additional experiments results using the Joint-EnKF$_{\text{OSA}}$. We showed that the proposed Dual-EnKF$_{\text{OSA}}$ slightly outperforms the Joint-EnKF$_{\text{OSA}}$ ($\sim 5\%$ better accuracy). We further reported the average ensemble spread, as estimated by the Joint-EnKF$_{\text{OSA}}$, for both states and parameters in Table 1.

**Detailed comments**

**C1.**  *L40: "have been proposed" instead of "has been proposed".*

**R1.**  Done. Thank you.

**C2.**  *L64: "was given" instead of "was carried"*

**R2.**  Done. Thank you.

**C3.**  *L99-L101: Rephrase.*

R3. We have rephrased the sentence which now reads as follows: "Our goal is to derive a new Dual-EnKF-like algorithm that retains the separate formulation of the state and parameters update steps, within a fully Bayesian framework."

**C4.**  *L109-L110: Change to: "(...) various experiment settings and observation scenarios."*

**R4.**   Done. Thank you.

**C5.**   *L122: this should be t(n-1) to t(n)?*

**R5.**   Based on our notation in system (1), our statement in L122 is correct: "$\mathcal{M}_n$ *is a nonlinear operator integrating the system state from time* $t_n$ *to* $t_{n+1}$".

**C6.**   *L203-L206: This was not found in Song et al. (2015, VZJ). There Dual EnKF performed worse, and only a rigorous Restart EnKF gave better results. Reformulate.*

**R6.**   We thank the referee for pointing this out. In fact, we have read the article by *Song et al. (2015)* and we were aware of this study prior to our submission. The authors tested the use of the Confirming Step EnKF, the restart EnKF and a modified variant of the restart EnKF. Unlike the Joint and the Dual EnKFs, these filters only update the parameters using a Kalman-type analysis. The state ensemble members, on the other hand, are obtained after integrating the model. Comparing the Confirming-Step-EnKF of *Wen and Chen (2006)*, which we cite in the manuscript, to that of *Song et al. (2015)* may not be that straightforward considering the differences between their models. On the one hand, *Wen and Chen (2006)* worked with a reservoir simulator and *Song et al. (2015)* used an unsaturated flow problem. It is possible that the efficiency of the Confirming-Step becomes more pronounced in nonlinear reservoir systems and strongly heterogenous subsurface formations, which is not the case in *Song et al. (2015)*. To avoid the confusion, we have removed the confirming-step EnKF reference from the discussion and reformulated the sentence.

**C7.**   *L377-L378: The pumping rate is unfortunately unrealistic low. It would have been better if the authors would have worked with a more realistic case.*

**R7.**   We fully understand the concern of the referee. The choice of the well pumping rates was based on contributions from the initial head values in addition to the present recharge. Starting from a uniform initial hydraulic head $h_0 = 15$ m, the recharge and the pumping rates eventually yield heterogeneous spatial distribution of $h$, varying between $\sim 13$ to 20 m, as shown in Figure 3. Increasing the pumping rates in our setup may, however, cause clear suction of the groundwater (negative head) and the dynamics will be mainly dominated by this forcing term. The considered rates create enough variability in space and time to test the assimilations schemes. Thank you.

**C8.**   *L587-L592: I do not see many differences and these are probably related to the initial conditions after the assimilation phase. Reconsider this text part.*

**R8.**   We are not sure what the referee exactly mean by the initial conditions at the end of assimilation. In fact, after the assimilation phase we simply continue the simulation in a prediction mode without assimilating any data. Thus, we don't tamper or impose any changes on the hydraulic head obtained after assimilation. In terms of the differences, we

[Figure]

Figure 1: Absolute bias resulting from the three assimilation schemes during the prediction phase at the control well.

disagree with the referee because the Joint-EnKF shows a clear overestimation of $h$ at the control well unlike the Dual-EnKF and more importantly the proposed Dual-EnKF$_\text{OSA}$. To illustrate, we evaluated the absolute bias during the prediction phase for the three schemes. We plot the resulting curves in Figure 1. As shown, the bias suggested by the proposed scheme is the smallest and around 0.5 m less than that of the Joint-EnKF. To support our argument, we have included this figure in the manuscript and we interpreted the results accordingly. Thank you.

C9. *Caption Figure 1. Change to: "The reference log-conductivity field was obtained (...)"*

R9. Done. Thank you.

C10. *Caption Figure 9. Why does AAE not decrease for joint EnKF and dual EnKF for small observation errors? Please comment.*

R10. The referee is raising an interesting point here. The performance of the Joint-EnKF and the Dual-EnKF clearly degrades when the observation error decreases, which might seem as counterintuitive. However, the errors only increase from 1.05 to 1.06 which is not significant. When the observational error decreases further to 0.1 m, the AAE of conductivity decreases to approximately 1.04. To test this further, we examined the performance of the filters with even smaller observation error, i.e., 0.05 m. We updated the figure in the revised manuscript, accordingly. We notice that the performance of the Joint-EnKF and the Dual-EnKF continue to improve reaching an AAE of around 1.01. Regarding the performance for measurement errors between 0.1 and 0.15, this could possibly be due to statistical fluctuations

related to this particular case and setup.

C11. *Caption Figure 9: "are obtained" is not correct.*

R11. Rephrased. Thank you.

C12. *Caption Figure 10. Why do you use lines in the figures? The legend is not clear.*

R12. The figure and its legend have been updated following the referee's comment.

**A Bayesian Consistent Dual Ensemble Kalman Filter for State-Parameter Estimation in Subsurface Hydrology**
by B. Ait-El-Fquih, M.E. Gharamti and I. Hoteit

**- Reply to Referee #2 -**

We would like to thank the referee for carefully reviewing our work and for his/her constructive comments. We have revised the manuscript taking into account all the referee comments and suggestions whose detailed replies are given below.

**Main concerns**

C1.  *A one-step-ahead smoothing based dual EnKF is presented in the manuscript. The authors compared results of the new method with standard joint and dual EnKF. As mentioned by the authors, Gharamti et al (2015) proposed a new EnKF method too by combining the one-step-ahead smoothing formulation (page 3). What will the result look like if these two one-step-ahead based EnKF methods are compared? Also the dual EnKF OSA needs to be better distinguished from the one by Gharamti et al (2015).*

R1.  We thank the referee for bringing this up. The referee raised two points:

- *Comparison of the filtering schemes:*

  We have examined the performance of the Joint-EnKF$_{\text{OSA}}$ of *Gharamti et al. (2015)* compared to the new filtering schemes in the first section of the results. We plot the resulting state and parameter estimates on top of the previous ones in Figure 4 and Table 3. The Joint-EnKF$_{\text{OSA}}$ is found more reliable during the early assimilation period but later and towards the end of assimilation, the proposed new scheme becomes clearly more accurate.

- *Differences between the algorithms:*

  The proposed Dual-EnKF$_{\text{OSA}}$ in this work results from the generic algorithm of Section 3.1 by applying two random sampling properties (see Appendix A) under the Gaussian assumption. At each assimilation step of the Dual-EnKF$_{\text{OSA}}$, the observed data are used three times through Kalman-like updates: twice in the smoothing step (one for smoothing the previous state as in Eq. (27) and one for updating the parameters as in Eq. (28)), and once in the "forecast" step to compute the analysis of the current state as in Eq. (35).

  The work in *Gharamti et al. (2015)* follows a similar approach and applies the same two random sampling properties on the generic algorithm of Section 3.1, but under the

following assumption (beside the Gaussian assumption)[2]:

$$p(\mathbf{x}_n|\mathbf{x}_{n-1}, \theta, \mathbf{y}_n) = p(\mathbf{x}_n|\mathbf{x}_{n-1}, \theta), \tag{2}$$

which is based on the fact that given the previous state, $\mathbf{x}_{n-1}$, and the parameters, $\theta$, the current state, $\mathbf{x}_n$, is independent of its observation, $\mathbf{y}_n$. Following this assumption, the "forecast" step of the generic algorithm which is composed of a Bayesian step (24) followed by a propagation step (23), reduces to the propagation step (23). As stated in page 445 of *Gharamti et al. (2015)*, the assumption (2) has been adopted to compute the analysis pdf, $p(\mathbf{x}_n|\mathbf{y}_{0:n})$, from the joint smoothing pdf, $p(\mathbf{x}_{n-1}, \theta|\mathbf{y}_{0:n})$, based on Eq. (9) (or Eq. (23) in our manuscript), by avoiding the use of the computationally demanding term $p(\mathbf{x}_n|\mathbf{x}_{n-1}, \theta, \mathbf{y}_n)$ and replace it by the more easily sampled state transition pdf, $p(\mathbf{x}_n|\mathbf{x}_{n-1}, \theta)$. Here, we propose a more efficient approach (see pages 10-11 and Appendix B) to directly sample from the analysis pdf without explicitly computing $p(\mathbf{x}_n|\mathbf{x}_{n-1}, \theta, \mathbf{y}_n)$ and without the need of any additional assumption.

Now, when applying the two random sampling properties (under the Gaussian assumption), the Joint-EnKF$_{\text{OSA}}$ in *Gharamti et al. (2015)* shares the same smoothing step as the Dual-EnKF$_{\text{OSA}}$, which, as mentioned above, involves two Kalman-like updates. However, in contrast with the Dual-EnKF$_{\text{OSA}}$, the Joint-EnKF$_{\text{OSA}}$ does not involve a Kalman-like update in the "forecast" step (because of the omission of the Bayesian step (24) from the generic algorithm). In terms of computational cost, these algorithms have almost the same cost as they require the same number of model runs; the only difference is one Kalman update for each member which is generally computationally not consequent compared to the cost of integrating the model. The differences between the proposed Dual-EnKF$_{\text{OSA}}$ and the Joint-EnKF$_{\text{OSA}}$ of *Gharamti et al. (2015)* were made clearer in lines 332-343 of the revised manuscript.

C2.    *Page 8. The authors talked about one-step-ahead smoothing function but did not explain what is this function used for? What role does it play in the new algorithm dual EnKF OSA and how does it work?*

R2.    We understand that the referee is referring to our statement (page 8, lines 245-246): "*...but also involves a (new) smoothing step that constraints the state with the future observation.*". This suggests that compared to the standard Dual-EnKF, the proposed Dual-EnKF$_{\text{OSA}}$ involves a new update step of the state using the future observation (hence the term "one-step-ahead (OSA) smoothing"). This smoothing "function" is given by Eq. (27), being a Kalman-like update of the state analysis members using the future observation. Line 317 of page 11 was updated to clarify the Kalman-like update character of the smoothing "function". Thank you.

C3.    *Page 11. The observation data are used three time in dual EnKF OSA rather than twice as in the dual EnKF. The authors thus concluded that it is in a fully consistent Bayesian*
* * *
[2]Refer to Eq. (16) in *Gharamti et al. (2015)*.

*formulation. Please clarify the relationship between them. Also please relate this with the comment on standard dual EnKF that the ensemble does not represent the forecast pdf (page 8 lines 231-233).*

R3.    We thank the referee for pointing this out. The proposed Dual-EnKF$_{\text{OSA}}$ is not a fully Bayesian algorithm because the observed data are used three times, but because this algorithm results from the generic (theoretically sound) Bayesian filtering algorithm presented in Section 3.1 by applying two random sampling properties (see Appendix A), and under the (commonly used) Gaussian assumption only. As stated in our reply R1 above, at each assimilation time step of the resulting Dual-EnKF$_{\text{OSA}}$, it turns out that the observed data are used three times through Kalman-like updates: one for smoothing the previous state (Eq. (27)), one for updating the parameters (Eq. (28)) and one for updating the current state (Eq. (35)). Accordingly, the use of the observed data three times is a consequence of the fully Bayesian character of the proposed scheme and not a cause. This was made clearer in the text below Eq. (35).

Regarding our statement on standard Dual-EnKF that the ensemble does not represent the forecast pdf and therefore it is a heuristic algorithm (page 8, lines 231-233), please note that this is related to the (Bayesian consistent) Joint-EnKF and not to the Dual-EnKF$_{\text{OSA}}$. Indeed, as one can see from Sections 2.2.1 and 2.2.2, the Joint-EnKF and the Dual-EnKF algorithms mainly differ in the state analysis step. The analysis members of the state, $\mathbf{x}_n^{a,(m)}$, are computed by the Joint-EnKF using a Kalman-like update of the forecast members, $\mathbf{x}_n^{f,(m)} = \mathcal{M}_{n-1}(\mathbf{x}_{n-1}^{a,(m)}, \theta_{|n-1}^{(m)})$ given in Eq. (5), while the Dual-EnKF computes $\mathbf{x}_n^{a,(m)}$ following the same mechanism but using the members $\widetilde{\mathbf{x}}_n^{f,(m)} = \mathcal{M}_{n-1}(\mathbf{x}_{n-1}^{a,(m)}, \theta_{|n}^{(m)})$ given in Eq. (12), instead of $\mathbf{x}_n^{f,(m)}$. However, in contrast with $\mathbf{x}_n^{f,(m)}$, the members $\widetilde{\mathbf{x}}_n^{f,(m)}$ are not samples from the forecast pdf (more generally, one does not know from which pdf $\widetilde{\mathbf{x}}_n^{f,(m)}$ are sampled) since they are obtained by integrating the model with the updated parameters, $\theta_{|n}^{(m)}$ (along with $\mathbf{x}_{n-1}^{a,(m)}$), instead of $\theta_{|n-1}^{(m)}$. Despite that, these are used in the analysis step as forecast members to compute the analysis members, $\mathbf{x}_n^{a,(m)}$, following the same Kalman-like update as in the Joint-EnKF, hence the heuristic nature of the Dual-EnKF.

Now, to relate that with the Dual-EnKF$_{\text{OSA}}$, one can easily see that this latter reduces to the Dual-EnKF in the particular case of a perfect model and in the absence of state smoothing (i.e., when Eq. (27) vanishes and $\mathbf{x}_{n-1}^{s,(m)} = \mathbf{x}_{n-1}^{a,(m)}$ in Eq. (33)). This was made clearer in page 11, lines 330-331 of the revised manuscript.

C4.    *Page 17 line 527-528 "The proposed dual EnKF OSA efficiently recovers the reference trajectory of MW2 and MW3". I think this statement is not so proper since the trajectory or trend of the reference is not captured well by any method, including the dual EnKF OSA. The reference is barely covered by the ensemble and the peak values are late in the ensemble at MW3. But I agree that the dual EnKF OSA works better than the other two methods.*

R4.    We have relaxed our argument here and rephrased the sentence accordingly. The sentence now reads as follows: "The proposed Dual-EnKF$_{\text{OSA}}$ performs fairly well, providing a reasonable recovery of the reference trajectory at MW2 and MW3.". Thank you.

C5. *Page 17 line 534 "We further examine ..... against the joint- and dual- EnKFs ...."* *But in Figure 7 only the results by dual-EnKF and dual- EnKF OSA are shown. The results by joint EnKF are not included.*

R5. We thank the referee for pointing this out. We have now included the results from the Joint-EnKF in Figure 7.

C6. *Page 18 line 576 "the dual- EnKF OSA tends to maintain a larger variance at the edges than the dual-EnKF, which in turn increases the impact of the observations". I have two questions on this: first, it is difficult to tell (Figure 10) the larger variance by dual-EnKF OSA. Their results look pretty similar. Secondly, why will the variance of hydraulic conductivity at the edge increase the impact of the observations (at 9 wells) which is not so near from the edges? Furthermore, the boundary conditions are either no-flow or constant head which have limited influence on observations. One suggestions on the color bar of the figure: the white color does not appear in the bar but occupy a lot of area in the Figure and the contours make the figure complicated.*

R6. We have improved the quality of the figure. Essentially, we removed the contour lines and kept the colouring. We further adjusted the colour bar to better emphasize the different behaviours of the schemes. In terms of the larger variance at the north boundary, we refer to the fact that an ensemble with larger spread (variance) is expected to fit more the observations using a Kalman-based update. However, if the ensemble spread is very small and the estimate is still far from the truth then the impact of assimilating future data would be minimal.

**Minor corrections**

C1. *Page 2 line 38, "Hendricks Franssen and Kinzelbach, 2009" instead of "Franssen and Kinzelbach, 2009". At the same time, please correct the item in the reference list (page 22).*

R1. Done. Thank you.

C2. *Page 5 line 150. "Let, for an ensemble ..., $\mathbf{r}$ denotes" should be "denote" instead of "denotes"?*

R2. Done, thank you.

C3. *Page 10. The equation 25 looks exactly the same as equation 5. Is this correct?*

R3. Yes, this is correct. However, mechanisms (5) and (25) are the same, but not their "outputs". In other words, the forecast members $\mathbf{x}_n^{f,(m)}$ resulting from (5) are not the same as those obtained by (25) even when starting from the same "input" $(\mathbf{x}_{n-1}^{a,(m)}, \theta_{|n-1}^{(m)})$ (which is very unlikely). This is because the noise $\eta_{n-1}^{(m)}$ in (5) are different than those of (25) even though they are sampled from the same Gaussian distribution $\mathcal{N}(\mathbf{0}, \mathbf{Q}_{n-1})$.

**C4**.   *Figure 1, the black crosses represent hard measurements according to the text on Line 425. But it can also be added here to avoid any confusion.*

**R4**.   Done. Thank you.

**C5**.   *Title of the subsection 5.5 "further assessment of the dual- EnKF OSA scheme" does not reflect the content. From the title we expect the result by dual- EnKF OSA only. But in fact it still compares the results of three methods. It could be changed to "prediction capability assessment" or something like this.*

**R5**.   We thank the referee for the comment. This title was changed as suggested.

[revised manuscript text omitted]

---

## Author Response (AR2)

Hydrology and Earth System Sciences (HESS) paper number hess-2015-544
**A Bayesian Consistent Dual Ensemble Kalman Filter for State-Parameter Estimation in Subsurface Hydrology**
by B. Ait-El-Fquih, M.E. Gharamti and I. Hoteit

Dear Professor Mauro Giudici,

First, we would like to thank you for giving us a chance to revise the manuscript. We also would like to thank you, as well as the referees, for all the constructive suggestions and comments you have made throughout the review process of our manuscript.

We have done our best to address your concerns as well as those of Referee #3. In particular, we have addressed your main suggestions, about improving the description of the novelty of our approach in the introduction section, and the discussion of the numerical (test case) results more highlighting how much they can be generalized and applicable to real case studies. We have further described in details the main concerns of Referee #3 as making clearer how the ensemble of members is initialized, among others.

All your comments and those of Referee #3 were carefully considered in the revised manuscript and detailed replies are given below.

We thank you again for all your editorial efforts.

Sincerely yours,

B. Ait-El-Fquih, M.E. Gharamti and I. Hoteit

Hydrology and Earth System Sciences (HESS) paper number hess-2015-544
**A Bayesian Consistent Dual Ensemble Kalman Filter for State-Parameter Estimation in Subsurface Hydrology**
by B. Ait-El-Fquih, M.E. Gharamti and I. Hoteit

**- Reply to Editor -**

**Main concerns**

**C1.** *The novelty of the paper is described in a more approriate way in the revised version of the manuscript (in sections 2 and 3), but it should be clear also from the introductory section.*

**R1.** We have followed the editor's suggestion to highlight more the novelty of our work in the introduction section. Thank you.

**C2.** *The numerical test has a purely "numerical" value, because the values of conductivity, recharge and well extractions are not realistic. Sorry to insist on this problem, but I am very convinced that it need to be fixed. At line 388, please erase "that is similar to a real-world application", or at least reformulate the sentence, because it seems to claim that the aquifer system under study is realistic, which is not the case.*
*Reply to comment 7 of the Referee #1 is not satisfactory. Above all, it should be clearly stated in the text that the conductivity field and the well pumping rates are not realistic. The average pumping rate for PW2 is about 2 nanoliters/second, if I performed correctly the units transformation! I am sure that you agree that this is a senseless value for real case studies. I understand that an increase of this value would cause drying of the aquifer, because the average conductivity is excessively (and unrealistically) small! Then, I expect you can add a discussion about the generalisation of the numerical results of this paper to real case studies. In other words, are your conclusion, still valid (not only from a qualitative point of view, but also from the quantitative point of view), for a real life application? Can you prove this?*

**R2.** We thank the editor for the comment. We have deleted "that is similar to a real-world application" and revised the corresponding sentence in the revised manuscript. We agree that the values assigned for conductivity and recharge rates are perhaps smaller than what is generally used in real-applications. This however should not affect the performance of the tested schemes. This is now acknowledged in the revised manuscript as suggested by the reviewer.

**Technical comments**

C1.  *Line 149: Please, provide an explicit definition of $\theta_{|n-1}^{(m)}$, in order to facilitate the reader. Please, rephrase "representing".*

R1.  $\theta_{|n-1}^{(m)}$ is now explicitly defined in the revised manuscript and the term "representing" replaced by "sampled from". Thank you.

C2.  *Line 158: Please, rephrase "the mean of...", possibly using the same symbol for expected value as used in equation (2).*

R2.  This sentence is now rephrased to: "The state forecast estimate, which is the mean of $p(\mathbf{x}_n|\mathbf{y}_{0:n-1})$ (i.e., $\mathbb{E}_{p(\mathbf{x}_n|\mathbf{y}_{0:n-1})}[\mathbf{x}_n]$), is taken as the empirical mean of the forecast ensemble, $\hat{\mathbf{x}}_n^f$.". Thank you.

C3.  *Line 183: If I understand correctly, the previous steps are the same as for Dual EnKF. If so, please, state it explicitly; if not, please clarify the difference.*

R3. Correct, the parameter subfilter of the Dual-EnKF computes the parameters ensemble members exactly as in the Joint-EnKF (eqs. (5), (6), (8)). This is now explicitly stated in line 181 of the revised manuscript. Thank you.

C4.  *Line 197: Substitute "geophysics" with "geophysical".*

R4.  Done. Thank you.

C5.  *Line 376: Information about the height of the impermeable bottom of the aquifer, which is necessary to compute b, is missing.*

R5.  Here we just assume the value of $b$ as 25 m. Again, we would like to mention that this is similar to what Bailey and Baù (2010) have considered in their assimilation study (as already mentioned in the manuscript).

C6.  *Line 377. Substitute "Neumann with no flow conditions" with "impermeable".*

R6.  Done. Thank you.

C7.  *Lines 385 & 386: Please, substitute "with a mean of 13 log(m/s), a variance of 1.5 log(m2/s2)," with "with a geometric mean of $10^{-13}$ m/s, a variance of $Y = \log K$ of 1.5,". I suggest you to use the same format throughout the whole paper, including tables and figures. Notice that the variance of $Y =$ is independent of the measurements units of $K$.*

R7.  Done. Thank you for the suggestion.

**C8.** *Line 488: Erase "s" from "deteriorates".*

**R8.** Done. Thank you.

**C9.** *Lines 539ff: Sorry, but this is a mere description of the figure; please, provide a discussion of the results, otherwise the paragraph is of poor interest.*

**R9.** We though that Figure 6 was well discussed in paragraph that contains line 539. We are not sure if the editor is referring to another paragraph/figure?

**C10.** *Line 580: Please, check the use of the expression "observation frequency" throughout the whole paper (including figures): "observation frequency" (whose dimension is inverse time) should be often replaced with "sampling period" (whose dimension is time).*

**R10.** Done. thank you.

**C11.** *Table 3: The number of decimal and significant figures is different for different values. Please, unify the format and check whether it is physically consistent.*

**R11.** Done. Thank you.

**C12.** *Figure 3: Numbers on the contour lines cannot be read.*

**R12.** We have improved the labelling of the figure. Thank you.

**C13.** *Figure 7: I see two scenarios only in the figures, not four. Please, check the plots.*

**R13.** We updated the caption of the figure. Thank you.

Hydrology and Earth System Sciences (HESS) paper number hess-2015-544
**A Bayesian Consistent Dual Ensemble Kalman Filter for State-Parameter Estimation in Subsurface Hydrology**
by B. Ait-El-Fquih, M.E. Gharamti and I. Hoteit

**- Reply to Referee #3 -**

We would like to thank the referee for carefully reviewing our work and for his/her constructive comments. We have revised the manuscript taking into account all the referee comments and suggestions. Our detailed replies are given below.

**Detailed comments**

**C1.**    *Lines 51-55: I suggest relaxing this sentence and limiting the ability to handle model structure errors to the cases presented in the cited reference (Hendricks Franssen and Kinzelbach, 1998). In fact the EnKF has been proven to be ineffective when other model structure errors such as, e.g., uncertain variogram model parameters, need to be taken into account (see, e.g., Jafarpour and Tarrahi, 2011: http://dx.doi.org/10.1029/2010WR009090 - "Assessing the performance of the ensemble Kalman filter for subsurface flow data integration under variogram uncertainty").*

**R1.**    We have revised the sentence following the reviewer's comment. Thank you.

**C2.**    *Lines 124-125: is it necessary to assume that parameters and state variables are independent? This doesn't seem to be realistic to me. You also state that there must be consistency between model parameters and initial hydraulic head fields (lines 439-442).*

**R2.**    The referee is right. This assumption is not needed and the sentence was removed accordingly. Thank you.

**C3.**    *Figure 1: I suggest adding the x and y axes labels with the corresponding units.*

**R3.**    We thank the referee for the suggestion. The units are already included in the upper-left corner of the domain. $x$ and $y$ labels are now described in the caption.

**C4.**    *Table 2 (caption): please correct "variorum" in "variogram".*

**R4.**    Done. Thank you.

**C5.**    *Lines 430-442: the procedure through which you initialize your ensemble members and the motivations for doing so are not clear to me. In more details:*

- *Lines 430-431: what is the "mean hydraulic head of the reference run solution"? Spatial mean? Temporal mean?*

- *line 432: "randomly select" from which set? Regardless of the fact that the same procedure has already been used by Gharamti et al. (2014), I suggest that all this initialization methodology should be explained more clearly in the manuscript.*

**R5.** The reference run is performed over a period of 1.5 years. Snapshots from the reference simulation are retained every time step; i.e., every 12 hours. Thus in total, we obtain 1095 hydraulic heads' snapshots. The temporal mean at every grid cell is then computed and used to carry out a new run with the perturbed forecast model over a 5 years period. This provided a set of 3650 head maps by saving the model outputs every time step. From these, we randomly select $N_e$ head maps and use it as the initial hydraulic head ensemble.

**C6.** *Equations 38-39: defined in this way, and consistently with the definition of vector $\mathbf{x}$ in equation (1), the two metrics AAE and AESP should refer to system states only. Subsequently, you employ AAE with reference to log-conductivies (e.g., in Figure 4, 5). Please consider revising this inconsistency.*

**R6.** We have generalized both metrics to account for both state and parameters. Thank you for the comment.

**C7.** *Line 476 and 478, caption of Table 3: I don't understand the reason for using the word "mean" before AESP. Shouldn't the AESP be an averaged quantity? Please consider dropping the word "mean" in "mean AESP". Otherwise state more clearly what you intend with the word "mean" in this context.*

**R7.** By "mean" we are referring to a temporal-mean AESP. The AESP is indeed a spatially-averaged quantity but in Table 3 we intended to show the overall mean from the entire 1.5-years assimilation. We have clarified this in the revised manuscript. Thank you.

**C8.** *Table 3: the results presented in Table 3 in my opinion are not adequately commented: the AESP indices related to log-conductivity do not increase with ensemble size as stated at lines 477. On the same line, it is not clear why the authors say "as expected".*

**R8.** The analysis of Table 3 results has been revised following the referee's comment. We agree with the referee, the ensemble spread for the state and the parameters behave differently as we change the size of the ensemble. The revised text reads as follows: "For all schemes, increasing the ensemble would increase the spread of the hydraulic head ensemble due to the natural variability of the considered subsurface system. In contrast, the AESP conductivity decreases as $N_e$ increases, probably because of the persistence nature of its prescribed dynamics.".

**C9.** *Lines 504-506: Does this mean that updating the model variables is more expensive*

*than running the forward model? Commonly, the updating step consists of a few algebraic equations that can be solved in a very short time, while the forward model run usually requires more time. Could you provide further details for this behavior?*

R9.    We totally agree with the referee. However, given the simplicity of our model, the cost of integrating a single member for 12 hours (one time step) is not very expensive compared to the update step. We further run all members in parallel on a 4-core machine. Yet, the forecast step is still more expensive than the analysis step. In realistic systems, the analysis step should be cheap compared to the forecast step. This was made clearer in the text.

C10.    *Section 5.4 (and also in many other further instances): you probably mean that the standard deviation of the measurement error is, e.g., 0.10 m, not the measurement error itself.*

R10.    The referee is right, we meant the standard deviation of the measurement error. We updated the text accordingly.

[revised manuscript text omitted]

---

## Author Response (AR3)

Hydrology and Earth System Sciences (HESS) paper number hess-2015-544
**A Bayesian Consistent Dual Ensemble Kalman Filter for State-Parameter Estimation in Subsurface Hydrology**
by B. Ait-El-Fquih, M.E. Gharamti and I. Hoteit

**- Reply to Editor -**

Dear Professor Mauro Giudici,

We would like to thank you for your decision of accepting our work for publication as well as for your careful reading of our manuscript. We have addressed all your comments and detailed replies are given below.

We thank you again for all your editorial efforts.

Sincerely yours,

B. Ait-El-Fquih, M.E. Gharamti and I. Hoteit

**Technical comments**

**C1.** *Line 55: Modify "some forms of model errors", as this expression is not informative.*

**R1.** We have made this expression more informative as suggested. Thank you.

**C2.** *Line 384: The saturated thickness is given by $b = h - z_{bot}$, where $z_{bot}$ is the height of the impermeable aquifer bottom. Details about $z_{bot}$ must be given: is it constant (horizontal aquifer bottom) or variable? If variable, please, specify how.*

**R2.** We agree with the editor. The height of the impermeable aquifer bottom, $z_{bot}$, is assumed constant in this case study and this is now mentioned in the text. Thank you.

**C3.** *Lines 393 & 394: The correction can be further improved. My (TeX) suggestion is: "with a geometric mean of $10^{-13}$ m/s, a variance of $Y = \log K$ equal to 1.5". The same holds for the caption of Figure 1.*

**R3.** Done, thank you.

**C4.** *Line 476: Substitute "frequency" with "period".*

**R4.** Done, thank you.

**C5.** *Line 495: I think it is preferable the expression "time-averaged AESP".*

**R5.** We have used "time-averaged AESP" as suggested. Thank you.

**C6.** *Line 541: Check the expression "as the frequency of observations in time decreases".*

**R6.** Thank you for pointing this out. We now use the expression "as observations are sampled less frequently in time".

**C7.** *Line 622: Substitute "observations sampling frequency" with "sampling period".*

**R7.** Done, thank you.

**C8.** *Table 2: Erase parentheses around the measurement units.*

**R8.** Done, thank you.

**C9.** *Figure 1: Substitute the heading of the colour scale with "$Y = \log K$, for $K$ in m/s". The same correction in Figure 8.*

**R9.** Done, thank you.

**C10.** *Figure 2: Modify the heading of the colour scale in analogy to what has been described here above for Figure 1. Also, check the values and maps, because recharge q should be given in m/s, since the dimension of q are [L/T] (see line 391).*

**R10.** Done, thank you.

**C11.** *Figure 3: The change to the labels are not sufficient to make them clearly visible and readable. Please, improve the label formats.*

**R11.** Done, thank you.

**C12.** *Figure 4: Erase "-water" from the y-axis title of the upper plot. Use "log K (K in m/s)" for the y-axis titles of the lower plot.*

**R12.** Done, thank you.

**C13.** *Figure 5: Substitute x-axis title with "Observation sampling period (days)".*

**R13.** Done, thank you.

**C14.** *Figure 7: Substitute the x-axes titles with "Time (months)".*

R14.    Done, thank you.